# Isoform-resolved transcriptome of the human preimplantation embryo

**Denis Torre** [1,14], **Nancy J. Francoeur** [2,14], **Yael Kalma**[3,14], **Ilana Gross Carmel**[3], **Betsaida S. Melo**[1,4], **Gintaras Deikus**[1,4,5], **Kimaada Allette**[1], **Ron Flohr**[6,7], **Maya Fridrikh**[1,4,5], **Konstantinos Vlachos**[8], **Kent Madrid**[1,4,5], **Hardik Shah**[1,4,5], **Ying-Chih Wang**[1,4,5], **Shwetha H. Sridhar**[1,4,5], **Melissa L. Smith**[9], **Efrat Eliyahu**[1,5], **Foad Azem**[3], **Hadar Amir**[3], **Yoav Mayshar**[10], **Ivan Marazzi**[11], **Ernesto Guccione** [12,13], **Eric Schadt** [1], **Dalit Ben-Yosef**[3,6,7,15] ✉ & **Robert Sebra** [1,4,5,13,15] ✉

Human preimplantation development involves extensive remodeling of RNA expression and splicing. However, its transcriptome has been compiled using short-read sequencing data, which fails to capture most full-length mRNAs. Here, we generate an isoform-resolved transcriptome of early human development by performing long- and short-read RNA sequencing on 73 embryos spanning the zygote to blastocyst stages. We identify 110,212 unannotated isoforms transcribed from known genes, including highly conserved protein-coding loci and key developmental regulators. We further identify 17,964 isoforms from 5,239 unannotated genes, which are largely non-coding, primate-specific, and highly associated with transposable elements. These isoforms are widely supported by the integration of published multi-omics datasets, including single-cell 8CLC and blastoid studies. Alternative splicing and gene co-expression network analyses further reveal that embryonic genome activation is associated with splicing disruption and transient upregulation of gene modules. Together, these findings show that the human embryo transcriptome is far more complex than currently known, and will act as a valuable resource to empower future studies exploring development.

During fertilization the human sperm and egg unite to form a primary totipotent cell, which undergoes sequential cleavages followed by lineage differentiation into >200 different cell types comprising the tissues and organs of the developing fetus. Occurring over roughly 7 days, these early phases of preimplantation development are regulated by extensive remodeling of gene expression, underlying one of the most complex and dynamic stages of development, and are considered one of the most fundamental paradigms in cell biology. Embryonic, pluripotent stem cells and organoids are used to mimic early stages of human development in vitro[1–3]; however, these are an approximation of the true molecular mechanisms occurring during development. Our understanding of human embryogenesis is largely inferred through developmental studies of model organisms such as zebrafish, mouse and primates[4–9]. Although this process is highly evolutionarily conserved, there are human-specific processes that have yet to be described due to the difficulty in studying human embryogenesis.

The advent of high-throughput next generation sequencing (NGS) has expanded our knowledge of the human transcriptome, facilitating gene expression profiling on a massive scale. However, current human gene annotation databases such as NCBI RefSeq[10], GENCODE[11] and Ensembl[12] are largely assembled using data derived from short-read RNA-sequencing (RNA-Seq) technologies. Due to limited read length, such technologies are inherently unable to capture the contiguous

sequence of most messenger RNAs (mRNAs)[13], often resulting in fragmented, incomplete, or incorrectly compressed isoform annotations. The development of single-molecule real-time sequencing (SMRT-seq) has overcome such limitations through sequencing full-length mRNA molecules up to 25 kb[14], eliminating the need for transcript assembly in silico. SMRT-seq has been applied to discover tens of thousands of previously unannotated isoforms in a variety of species including humans, mice, other vertebrates, invertebrates and plants[13,15–19]. Many of these studies integrate additional short-read RNA-Seq data to improve the power of isoform expression quantification[20]. This approach was recently successfully applied to study the mouse preimplantation embryo, leading to the identification of thousands of unannotated isoforms transcribed from both known and novel gene loci[15]. However, human preimplantation embryos have been characterized using short read data from a limited number of studies to date[21–27], motivating the need for an isoform-resolved approach to comprehensively profile RNA expression and splicing during these critical stages of development. Indeed, alternative splicing (AS) has been demonstrated to be critical for proper oogenesis and pre-implantation embryonic development[28,29]. Similarly, in vitro studies demonstrated transcriptome-wide AS dynamics that are key in the establishment and exit from pluripotency[30–33] reflecting on the importance of splicing in the inner cell mass (ICM) of blastocysts.

Combining full-length isoform structures uncovered by SMRT-seq with the high read depth of RNA-Seq, we present the first isoform-resolved catalog of transcriptional changes across early time points in human embryogenesis using high quality in vitro fertilization (IVF) embryos spanning six preimplantation stages from the zygote to the blastocyst. These embryos were donated for research by patients after completing their family fertility plan and following informed consent. We subsequently leveraged the data to better characterize isoform open reading frames (ORFs), repetitive element content, evolutionary conservation; validated these isoforms by integrating published multi-omics datasets generated from human and non-human primate embryos, in-vitro human blastoids and 8-cell like cells (8CLCs), as well as fetal and adult tissues; and further explored dynamics of differential gene expression and alternative splicing over preimplantation developmental stage transitions. Our data reveals 110,212 unannotated alternative splice variants of known genes and 17,964 unannotated isoforms transcribed from 5239 novel gene loci, which suggests that the human transcriptome is indeed far more complex and dynamic than current annotations indicate, and will thus serve as a valuable resource for developmental studies to further explore the role of critical genes in development and disease.

## Results

### Long-read RNA-Seq identifies unannotated isoforms in human embryos

We generated RNA sequencing data from 73 human embryos across six pre-implantation stages: 13 zygotes (1C), 13 2-cell (2C) embryos, 16 4-cell (4C) embryos, 15 8-cell (8C) embryos, 3 morulae and 13 blastocysts (Fig. 1A, Supplementary Data 1). mRNA was extracted from each embryo, converted to cDNA, and sequenced using high-throughput Illumina RNA-Seq and SMRT-seq, generating a total of $4.3 \times 10^9$ short reads and 10,139,308 full-length non-concatemer (FLNC) long reads with an average length of 1332 bp (Supplementary Fig. 1A–C).

The long and short RNA-Seq reads were then cohesively analyzed to generate a reference transcriptome using an integrative analysis pipeline with multiple state-of-the-art computational tools (Supplementary Fig. 1D–F, see Methods). Isoforms were classified into five structural categories based on their similarity to known isoform annotations (Fig. 1B): "known", if the isoform is an exact match of a known transcript model; "novel in catalog" (NIC), if the isoform consists of a novel combination of known splice donors and acceptors;

"novel not in catalog" (NNC), if the isoform contains at least one novel splice donor or acceptor; "antisense", if the isoform is transcribed from a novel gene which overlaps an existing gene but is oriented in the opposite direction (novel antisense gene); and "intergenic", if the isoform is transcribed from a novel gene which does not overlap with any known transcript model (novel intergenic gene). Two additional isoform classes were also defined (incomplete splice match, if the isoform is a partial match of a known transcript model, and novel mono-exonic isoforms), but not included in the final transcriptome as these are often artifacts of sequencing and/or products of transcript degradation[34]. We identified a total of 213,873 isoforms, supported by junction-spanning short RNA-Seq reads, which are either known or newly identified from the SMRT-seq data. Most of these isoforms are transcribed from known genes: 85,697 (40.1%) were classified as known, 30,988 (14.5%) as NIC, and 79,224 (37.0%) as NNC (Fig. 1C). Another 17,964 isoforms were identified from unannotated loci: 9457 isoforms (4.4%) transcribed from 2466 novel antisense genes, and 8507 (4.0%) from 2773 novel intergenic genes. We found that known genes transcribe an average of 4.6 known and 10.2 novel isoforms; by contrast, novel antisense and intergenic genes displayed lower averages of 3.8 and 3.1 isoforms respectively (Supplementary Fig. 1G). NIC isoforms have an average of 8.6 exons, the highest of all isoform classes, followed by 7 for NNC and 6.5 for known isoforms; while novel antisense and intergenic isoforms had averages of 3.5 and 3.1 exons respectively (Supplementary Fig. 1H). Over 99% of isoforms across all classes exclusively use canonical splice donor-acceptor sites; the only exception being NNC, 3% of which contain at least one noncanonical splice junction, potentially arising from yet uncharacterized atypical splicing mechanisms[35] (Supplementary Fig. 1I). Thus, our data suggests that the human genome is heavily transcribed during early stages of development, with a much higher splicing diversity than currently annotated.

### Characterizing biological properties of novel isoforms

To further characterize the novel isoforms, we predicted multiple biological properties from their nucleotide sequence. First, we assessed the protein coding potential of each isoform using CPAT[36]. Coding probability was positively associated with isoform length, with isoforms transcribed from known genes predicted to have significantly higher coding probability than ones transcribed from novel genes (see Fig. 1D and Supplementary Fig. 2A, $p < 2.2 \times 10^{-16}$, rho = 0.54, Spearman's correlation coefficient). Interestingly, NIC isoforms were predicted to have the highest coding probability among isoform classes, followed by known, NNC, antisense and intergenic. Indeed, these isoforms have a longer median length, a higher number of exons, and are transcribed from genes which are significantly more protein-coding than others (Supplementary Fig. 2B). Despite being transcribed by a highly overlapping set of genes, NNC isoforms are instead slightly more associated with genes with lower coding probability (Supplementary Fig. 2C), suggesting that these loci may contain a large amount of previously uncharacterized splice sites in early embryonic stages. Next, we applied PfamScan[37] to identify protein domains contained within the ORF sequences of the predicted coding isoforms. Most isoforms transcribed from known genes were predicted to generate proteins with at least one known domain (Fig. 1E), the likeliest class being NIC (63.4%), followed by known (53.9%) and NNC (49%). On the contrary, only 0.8% antisense isoforms ($n = 78$) and 0.5% intergenic isoforms ($n = 43$) were predicted to do so. Notably, the latter are significantly overrepresented for reverse transcriptase (RVT 1), viral coat polyprotein (TLV coat) and transposase domains, which mediate retrotransposon replication and insertion, suggesting that these elements may contribute to the generation and function of novel isoforms. Indeed, transposable elements (TEs) play a fundamental role in regulating cell development and differentiation,

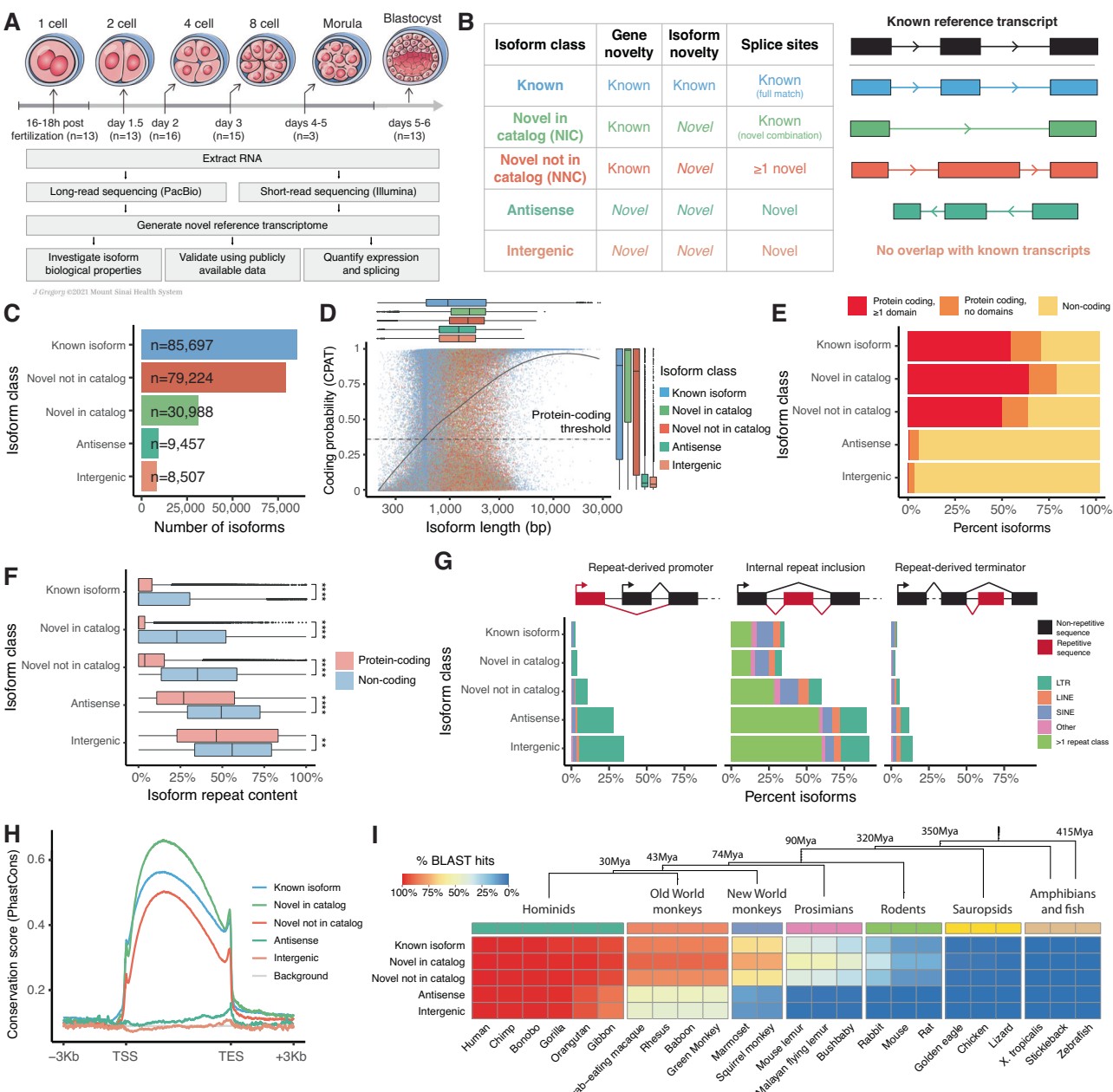

**Fig. 1 | Generation and functional characterization of the isoform-resolved human embryo transcriptome. A** Overview of the embryonic developmental stages and the sequencing approach (illustration by Jill Gregory). **B** Schematic representation of the isoform structural classes defined from long-read RNA-Seq data. **C** Number of isoforms in the novel human embryo transcriptome for each structural class. **D** Scatter plot displaying isoform length and predicted coding probability for each isoform, colored by isoform class. Residual boxplots display the distributions of isoform length and coding probability along the X and Y axes respectively. **E** Bar plot displaying the percentage of isoforms in each class based on their predicted protein-coding status, and the presence of known protein domains in the encoded peptide. **F** Box plots displaying the relative repeat content of isoforms in each structural class, grouped by predicted protein-coding status. For known isoforms, $n = 59{,}517$ protein-coding and $n = 26{,}180$ non-coding; novel in catalog, $n = 24{,}018$ protein-coding and $n = 6970$ non-coding; novel not in catalog, $n = 49{,}826$ protein-coding and $n = 29{,}398$ non-coding; antisense, $n = 550$ protein-coding and $n = 8907$ non-coding; intergenic, $n = 289$ protein-coding and $n = 8218$

non-coding. $p < 2{\times}10^{16}$ for known, novel in catalog, novel not in catalong and anti-sense isoforms, $p = 0.0062$ for intergenic isoforms. *P*-values were calculated using unpaired two-sided Wilcoxon Rank Sum test, Benjamini-Hochberg correction. **G** Bar plots displaying the relative abundance of repetitive elements acting as alternative promoters, internal exon elements, or terminators for each isoform class, grouped by repeat class. **H** Average base-wise conservation scores (Phast-Cons100way) across exons and ±3 kb of each isoform, grouped by isoform class. **I** Evolutionary conservation of transcripts across multiple vertebrates, in relation to the phylogenetic tree. The heatmap displays the percentage of conserved isoforms in each structural class compared to different vertebrate genomes, determined using BLAST. The phylogenetic tree displays evolutionary divergence of selected vertebrate groups from hominids. For the box plot in **F**, box limits extend from the 25th to 75th percentile, while the middle line represents the median. Whiskers extend to the largest value no further than 1.5 times the inter-quartile range (IQR) from each box hinge. Points beyond the whiskers are outliers. Source data are provided as a Source Data file.

initiating stage-specific transcription and providing promoter modules to both embryonic and somatic tissues[38–41].

To further investigate the presence of TEs in our transcriptome, we applied RepeatMasker[42] to scan the isoforms for repetitive elements. Known isoforms are the least repetitive class, with most isoforms containing negligible repetitive content as a fraction of their total length (Fig. 1F). However, repetitive elements were significantly more included in non-coding isoforms than in their protein-coding counterparts, consistent with their known association with long non-coding RNA (lncRNA) transcription[43]. Similar patterns were observed across all novel isoform categories, but with significantly higher levels of repetitive element inclusion, especially for non-coding intergenic isoforms. To elucidate the biological role of these widespread integration events, we categorized them by repetitive element class and the relative location of integration within the isoform (Fig. 1G). Most repeat-derived promoters are driven by TEs containing long terminal repeats (LTRs). LTRs serve as promoters for over a quarter of antisense and intergenic isoforms ($n = 2267$ and 2525 respectively), as well as to 526 known, 947 NIC, and 6086 NNC isoforms. These include HERVH-int, THE1D and MLT2A1, which have been previously shown to form chimeric isoforms with known genes[39,44,45]. Repetitive elements are abundantly integrated within isoforms across all categories, predominantly among novel antisense and intergenic classes, where they may alter RNA processing mechanisms or contribute binding sites for RNA-binding proteins[46]. Repeat-derived transcriptional end sites (TESs) are the least abundant category, most commonly occurring in novel antisense and intergenic genes where LTRs, SINEs and LINEs account for roughly 11% of terminator sequences. These results show widespread evidence of repetitive element integration within isoforms, which has thus far been lacking in current annotations, likely due to the technical limitations of short-read sequencing.

We next investigated isoform evolutionary conservation at the sequence level, which may be used to estimate divergence time of novel transcripts, and to prioritize elements with putative conserved biological function across species (though it is important to note that a lack of conservation does not imply lack of function[47,48]). We measured isoform conservation using PhastCons[49], which estimates base-wise conservation from multiple sequence alignment of the human genome against 99 other vertebrate species (Fig. 1H). Novel antisense and intergenic isoforms displayed the lowest evolutionary conservation scores of all classes, but were still significantly more conserved than intergenic background (Supplementary Fig. 2D). By contrast, isoforms transcribed from known genes displayed significantly higher conservation scores, with NIC being the most conserved category. Indeed, these isoforms are disproportionately transcribed from highly evolutionarily conserved genes, and their sequences lie within annotated splice sites that are more highly conserved than their NNC counterparts (Supplementary Fig. 2E–H). All isoform classes also displayed peaks of evolutionary conservation at their TESs, indicating the presence of conserved DNA elements responsible for driving transcriptional termination[50]; indeed, poly(A) motifs were identified close to the 3′ end for isoforms across all categories (Supplementary Fig. 2I, J). Conservation peaks were also observed at the TSSs of isoforms transcribed from known genes but less at novel genes, likely because the TSSs of the latter often lie within human-specific repetitive elements. To further assess conservation at the species level, we scanned isoform sequences against multiple vertebrate genomes using BLAST[51] (Fig. 1I). Predictably, over 99% of all isoforms were classified as hits (>100 bp sequence match with >95% identity) in chimp and bonobo, two of the most closely related primates to humans. However, in species with greater evolutionary distance to humans, the conservation of isoforms transcribed from novel genes decreased more rapidly compared to isoforms transcribed from known genes. In the macaque, only 47.6% antisense and 42.8% intergenic isoforms are classified as hits, compared to 85.3% of known isoforms; in the marmoset, only 12.1%

antisense and 9.7% intergenic isoforms are classified as hits, compared to 65.5% of known. In the mouse, one of the most common models to study mammalian embryogenesis, only 0.3% antisense and 0.1% intergenic isoforms were classified as hits, compared to 10.4% of known isoforms. While the number of known isoforms concordant with the mouse genome rapidly increases upon lowering the minimum sequence identity threshold, the number of novel antisense and intergenic isoforms in common remains consistently low, further supporting the novelty of these transcriptional events (Supplementary Fig. 2K). Thus, these results suggest that common rodent models for developmental studies are likely unable to recapitulate significant components of primate embryonic development, particularly for non-coding transcripts. Nonetheless, it is also possible that some of these isoforms represent by-products of transcriptional events occurring during early embryonic development. Additionally, secondary structures could also impart evolutionary conserved functions that are not apparent when considering primary sequence alone. Full results for the protein-coding probability, protein domain, repeat element content and evolutionary conservation analyses for each isoform can be found at Supplementary Data 2–5. Results are also available in the accompanying resource website and browser, https://denis-torre.github.io/embryo-transcriptome/, which allows users to interactively explore the splicing patterns and predicted biological function for every isoform in the isoform-resolved reference transcriptome, and freely download all relevant data files for further reanalysis.

## Multi-omic validation of long-read isoforms

To confirm the validity of the isoforms reported herein, we integrated multiple datasets from independently published transcriptomic and epigenomic studies conducted on early human embryos (Fig. 2A, Supplementary Data 6). The transcriptomic datasets were processed to validate the expression of isoforms, while the epigenomic datasets were processed to assess chromatin state at unannotated TSSs throughout development.

First, we integrated three short-read RNA-Seq studies profiling human embryos at comparable time points (Yan et al.[21], Xue et al.[23], Liu et al.[22]) and investigated the number of isoforms across classes that are fully supported by spliced short reads across all junctions. Most isoforms in the updated transcriptome are supported by short RNA-Seq reads across all three published transcriptomic datasets analyzed: 74.2% known isoforms, 69.7% NIC, 71.4% NNC, 77% antisense, and 76.5% intergenic (Fig. 2B). These values are even higher when counting support in at least one dataset: 99.2% known isoforms, 98.1% NIC, 98.3% NNC, 99.3% antisense, and 99.1% intergenic. Despite this concordance, the contiguous sequence of these isoforms was not known at the time these datasets were published, underscoring the utility of our isoform-resolved transcriptome for retrospective analyses. While Yan et al. carried out de novo transcript assembly, this approach only leveraged short-read RNA-Seq data and was thus unable to capture the complete isoform structures captured herein. We further leveraged these datasets to assess the expression dynamics of isoforms across developmental stages (Fig. 2C), alongside data from three additional studies that span subsets of this timeline: a single-cell RNA-Seq dataset spanning human embryos between E3-E7 (Petropoulos et al.[24]) and two datasets containing a large number of oocyte and 1C samples (Asami et al.[25] and Töhönen et al.[26]). Most novel isoforms are broadly expressed between the 1C and 8C embryonic stages across all datasets, and subsequently downregulated in the morula and blastocyst (Supplementary Fig. 3A). This is especially evident for novel antisense and intergenic isoforms, which reach a peak of 97% detection at the 4C stage in our short-read RNA-Seq samples (compared to 87% of known isoforms), but only about 48% in the blastocyst (compared to 89% of known isoforms). Similar patterns were observed in the publicly available data, albeit at lower detection rates, which may be partly explained by the lower sequencing depth of these studies

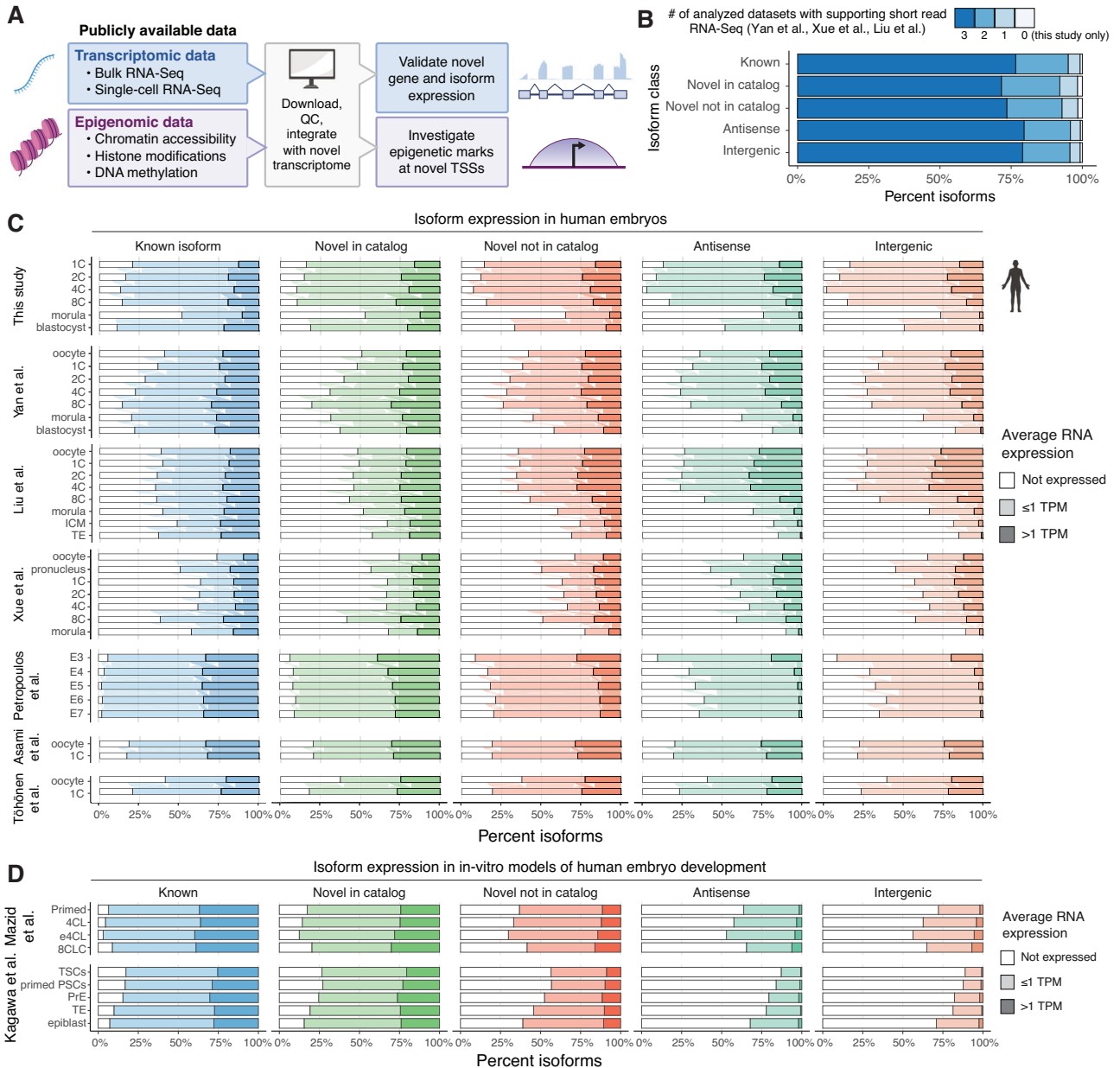

**Fig. 2 | Novel isoforms and genes are broadly expressed in early preimplantation stages.** **A** Overview of the data types integrated and approach to validate the novel isoform-resolved transcriptome. **B** Percentage of isoforms in each class, grouped by the number of integrated short-read RNA-Seq datasets in which they are expressed (Yan et al., Liu et al., Xue et al.). **C** Percentage of isoforms in each developmental stage, grouped by isoform class and average expression level across short-read RNA-Seq datasets profiling human preimplantation embryos and oocytes (TPM – Transcript Per Million). **D** As above, but displaying data from single-cell short-read RNA-Seq datasets (SmartSeq2) profiling in-vitro models of human preimplantation development (8CLCs and blastoids). Source data are provided as a Source Data file. Icons in Fig. 2A, C were created with BioRender.com.

(Supplementary Data 6). We found that many novel isoforms are also detected in human oocytes, indicating these are already expressed prior to fertilization and may include previously uncharacterized maternal transcripts.

We also integrated single-cell RNA-Seq data from two recent studies characterizing emerging platforms to study early human development in vitro: 8-cell-like cells (8CLCs), which mimic the human embryo 8C phase and are derived from human pluripotent stem cells (hPSCs, Mazid et al.[52]) and blastoids, in vitro hPSC-derived structures which mimic the human blastocyst (Kagawa et al.[53]). Notably, we found increased expression of all novel isoform classes in cells during the primed PSC to 8CLC conversion, and a similar increase in expression of such isoforms in cells that are part of blastoid structures when

compared to primed PSCs (Fig. 2D). The expression of novel isoforms alone was shown to effectively separate developmental stages and cell types from the three integrated single-cell RNA-Seq studies (Supplementary Fig. 3B). Taken together, these data show that the novel isoforms reported are widely supported across published studies spanning multiple modalities as well as newly developed in-vitro models.

We further integrated chromatin accessibility, H3K4me3, and H3K27ac data from two independent studies (Liu et al.[22], and Xia et al.[54]), to assess whether the novel isoforms are associated with epigenetic marks of active transcription[55–57]. We first defined four TSS classes: known TSSs, novel TSSs of known genes, and TSSs of novel antisense and intergenic genes respectively. Roughly 90% of novel

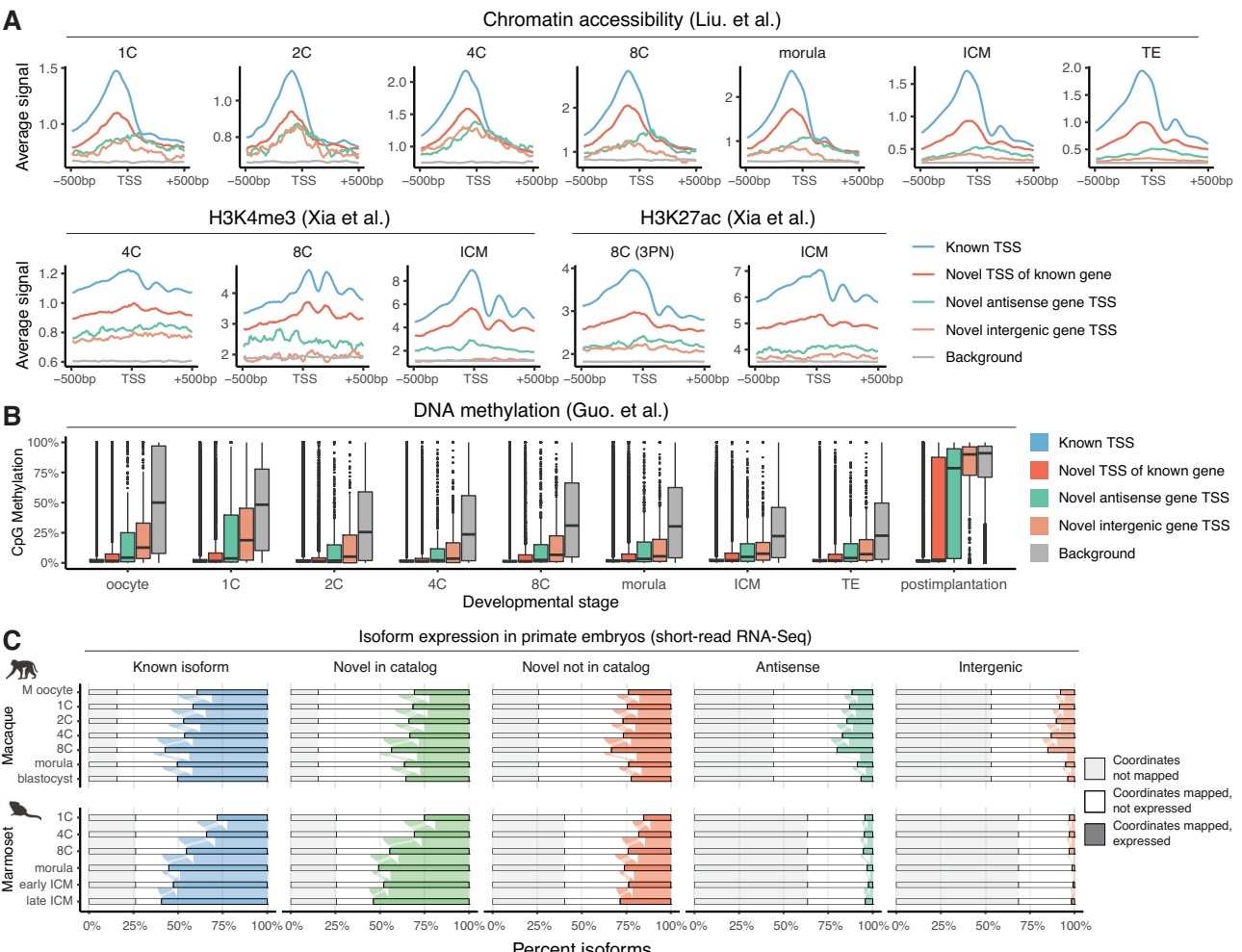

**Fig. 3 | Novel isoforms are supported by orthogonal epigenomic and non-human primate embryo transcriptomic data. A** Integration of public datasets with the newly identified transcriptome demonstrate epigenetic marks for active transcription across developmental stages. Data are normalized ATAC-Seq and CUT&RUN signal within ±500 bp of TSSs ($n$ = 78,217 known TSSs, 21,016 novel TSSs of known genes, 3589 novel antisense gene TSSs, 3814 novel intergenic gene TSSs, 106,636 background sequences). **B** Distribution of CpG methylation at TSSs from the novel transcriptome across developmental stages (percentages within ±500 bp of each site, TSSs defined in **A**). **C** Percentage of isoforms in each class, grouped according to their mapping status to the macaque and marmoset genomes and their expression in corresponding preimplantation short-read RNA-Seq datasets. For the box plot in **B**, box limits extend from the 25th to 75th percentile, while the middle line represents the median. Whiskers extend to the largest value no further than 1.5 times the inter-quartile range (IQR) from each box hinge. Points beyond the whiskers are outliers. Icons in **C** were created with BioRender.com. Source data are provided as a Source Data file.

TSSs of known genes and 65% of novel TSSs of antisense genes lie within known gene bodies or in proximity of known promoter regions; by contrast, about 95% of novel intergenic TSSs lie within distal intergenic space (Supplementary Fig. 3C). For each category and epigenetic dataset, we calculated average normalized pileups in genomic windows ±500 bp from each TSS, comparing them to intergenic background regions (Fig. 3A and Supplementary Fig. 3D). Known TSSs displayed the highest levels of chromatin accessibility, H3K4me3, and H3K27ac across all developmental stages and datasets, followed closely by novel TSSs of known genes. By contrast, novel antisense and intergenic gene TSSs displayed lower levels of these marks, but were still significantly higher than background at each stage ($p < 2.2 \times 10^{-16}$, Wilcoxon rank-sum test). Given that many such genes are already detected in oocytes, this result suggests that some of these are maternally inherited and not actively transcribed in early preimplantation development. Consistent with observed upregulation patterns, TSSs of novel antisense and intergenic genes were most accessible and highly associated with H3K4me3 and H3K27ac between the 4C and 8C stages, decreasing to near background levels in the blastocyst inner cell mass (ICM). Similar patterns

were observed for novel antisense TSSs, but with slightly higher levels, likely in part due to their proximity to actively transcribed known genes. We additionally integrated an independent dataset profiling DNA methylation in early human embryos (Guo et al.[58]), which plays a key role in transcriptional repression[59–62] (Fig. 3B). Known TSSs were the least methylated category, harboring close to 0% mCpGs within ±500 bp of each site across all profiled stages (1C to postimplantation). By contrast, novel TSSs displayed higher mCpG levels, but still significantly lower than background in all pre-implantation samples ($p < 2.2 \times 10^{-16}$, Wilcoxon rank-sum test). Interestingly, we observed pronounced hypermethylation in the post-implantation sample disproportionately affecting novel TSSs, particularly of novel genes. For example, the percentage of hyper-methylated novel intergenic TSSs (≥50% mCpGs within ±500 bp) increases from 7.1% in the TE to 85.9% in the post-implantation stage, while the corresponding values for known TSSs are 3.8% and 15.7%, respectively. This is consistent with our observation that TSSs of many novel genes lie within transposable elements, which are known to be broadly methylated and epigenetically silenced in somatic tissues[63,64]. Together, these results indicate that novel isoforms are

widely expressed and associated with epigenetic marks of active transcription in early preimplantation stages, with many novel genes likely undergoing transcriptional silencing by DNA methylation following embryo implantation.

Next, we sought to investigate whether the novel isoforms and genes are also expressed in non-human primate embryos. To achieve this, we analyzed RNA-Seq data from embryonic studies of the rhesus macaque (*Macaca mulatta*)[65] and the common marmoset (*Callithrix jacchus*)[66]. (Supplementary Data 6). First, we mapped the genomic coordinates of both known and novel isoforms from the human genome to the respective primate genomes using liftOver[67], discarding isoform models with failed or incomplete mapping from further analysis to improve accuracy. We then estimated the expression of the fully mapped isoforms using short-read RNA-Seq data across primate preimplantation stages. Predictably, known isoforms have the highest degree of mapping to the primate genomes, and are broadly detected across developmental stages (Fig. 3C, Supplementary Fig. 3E). Novel isoforms of known genes were also widely mapped and expressed in both species, suggesting that many previously undetected alternative splicing events are conserved in non-human primates. By contrast, novel antisense and intergenic isoforms displayed the lowest levels of conservation and expression in both primates. For example, only 24% of novel intergenic isoforms were mapped and detected in macaque preimplantation embryos, and only 7% in the marmoset (Supplementary Fig. 3F). Nonetheless, the mapped isoforms displayed similar expression patterns to the human, reaching highest detection levels between the 4C and 8C stages and subsequently undergoing downregulation. Thus, the human transcriptome described in this study includes both human-specific and evolutionarily conserved novel isoforms and genes supported by various independent transcriptomic and epigenomic studies.

Lastly, to investigate whether novel isoforms and genes are expressed in more mature human fetal and adult tissues, we integrated published short-read RNA-Seq data generated from 7 different organs at multiple time points of human development spanning week 4 postconception through adulthood[68]. We found that NIC are the most highly detected class, with up to 60% isoforms detected across multiple fetal tissues, followed by NNC, with a detection rate of around 20% (Supplementary Fig. 3G). These levels tend to decrease throughout development, with fewer such isoforms observed in adult samples. By contrast, novel antisense and intergenic isoforms were less significantly detected, even in fetal tissues, suggesting that these transcriptional events are primarily restricted to earlier developmental stages.

## Known developmental genes transcribe novel isoforms

Gene and isoform expression dynamics were further examined across human preimplantation stages. Principal Component Analysis (PCA) of gene expression levels across our short-read RNA-Seq samples revealed strong separation between early developmental time points (1C, 2C and 4C) and later stages (8C, morula and blastocyst), consistent with the known timing of major embryonic genome activation (EGA, Fig. 4A)[69]. We then sought to measure how strongly novel isoforms contribute to gene expression levels across developmental stages. To estimate this, we identified genes that are confidently expressed at each developmental stage using our polyA+ short-read RNA-Seq data, and then calculated the average percentage of such reads that are predicted to derive from novel isoforms for each gene and stage (Fig. 4B). The 4C and 8C stages displayed the highest degree of isoform novelty, with over 50% of expressed genes per stage predicted to be predominantly transcribed as novel isoforms in their polyadenylated fraction, while this value significantly decreases to about 25% in the blastocyst ($p < 2.2 \times 10^{-16}$, Wilcoxon rank-sum test). This suggests that genes expressed at earlier developmental stages are poorly annotated, likely due to the difficulty in establishing experimental models for such

early time points especially when limited to short-read RNA-Seq data. We further assessed isoform temporal expression dynamics by performing a differential expression analysis using sleuth[70]. We found thousands of differentially expressed isoforms, with a peak of differential expression taking place during the 4C to 8C transition, coinciding with EGA (Fig. 4C). Novel antisense and intergenic isoforms are significantly enriched among downregulated isoforms at each developmental transition starting from the 8C stage, confirming that these are broadly downregulated beyond EGA (Supplementary Fig. 4A). This pattern is also evident at the epigenetic level, with novel antisense and intergenic isoform TSSs displaying lower degrees of chromatin accessibility, H3K4me3 and H3K27ac deposition from the 8C stage onwards (Supplementary Fig. 4B).

To determine whether known developmental genes transcribe novel isoforms, we next focused on a set of 74 genes which have been previously reported as regulating development[66, 71–82] and undergo statistically significant changes in expression throughout preimplantation stages (Fig. 4D, Supplementary Data 7). These include early markers such as *DNMT1*, *PADI6* and *FOXO3*; pluripotency markers such as *SOX2*, *NANOG* and *OCT4/POU5F1* and blastocyst markers including *GATA3*, *CD24* and *KLF6*. Most of these genes were found to transcribe multiple novel isoforms, particularly at earlier preimplantation stages, including both non-coding and protein-coding RNAs with varying ORF lengths, predicted protein domains, and chimeric TE-gene isoforms. For example, we found 156 novel isoforms of *DNMT1*, a DNA methyltransferase involved in the maintenance of methylation imprints in preimplantation embryos[83]. These include a novel major TSS used across all preimplantation stages, two novel exons predicted to be in frame and thus contribute to the isoform ORF sequence, and multiple short isoforms predicted to produce N- and C-terminal truncated proteins containing diverse combinations of its protein domains, including ones conferring DNA-binding and methyltransferase function[84] (Fig. 4E, Supplementary Fig. 4C). We also identified 21 novel isoforms for *PADI6*, an evolutionarily conserved maternal factor which catalyzes protein deimination[85]. These include several short isoforms containing novel LINE-derived TESs, which are predicted to generate shorter C-terminal truncated peptides with fewer protein-arginine deiminase (PAD) domains (Supplementary Fig. 4D). We also identified 8 novel isoforms for *FOXO3*, a transcription factor which regulates mouse preimplantation development[86], including a novel TSS and an in-frame exon (Supplementary Fig. 4E). These results indicate that the human preimplantation transcriptome is far more complex than currently annotated, even for many well-studied developmental genes.

## Novel isoforms are transiently included during EGA

Having observed widespread expression of novel isoforms across developmental genes, we sought to further leverage this data to explore the patterns of alternative splicing (AS) over time. First, we used SUPPA2[87] to measure the relative abundance of seven major types of AS events (Fig. 5A). We identified hundreds of statistically significant AS events taking place across developmental stages, with the highest splicing diversity taking place in the morula-to-blastocyst and 4C-to-8C transitions respectively (Fig. 5B). Genes undergoing AS were found to be significantly enriched for pathways including mRNA processing, splicing and translation (Supplementary Fig. 5A). Interestingly, the 4C-to-8C transition displayed a significant increase of exon skipping and intron retention events, which are typically associated with splicing disruption[88–90].

We next performed an isoform switching analysis that included integrated predictions on the biological properties of AS to better understand the effect that such events have on gene function. Similar to what was observed for AS events (Fig. 5B), we found peaks of isoform switching at the morula-to-blastocyst and 4C-to-8C transitions, with nearly 500 and 200 isoform switching events respectively

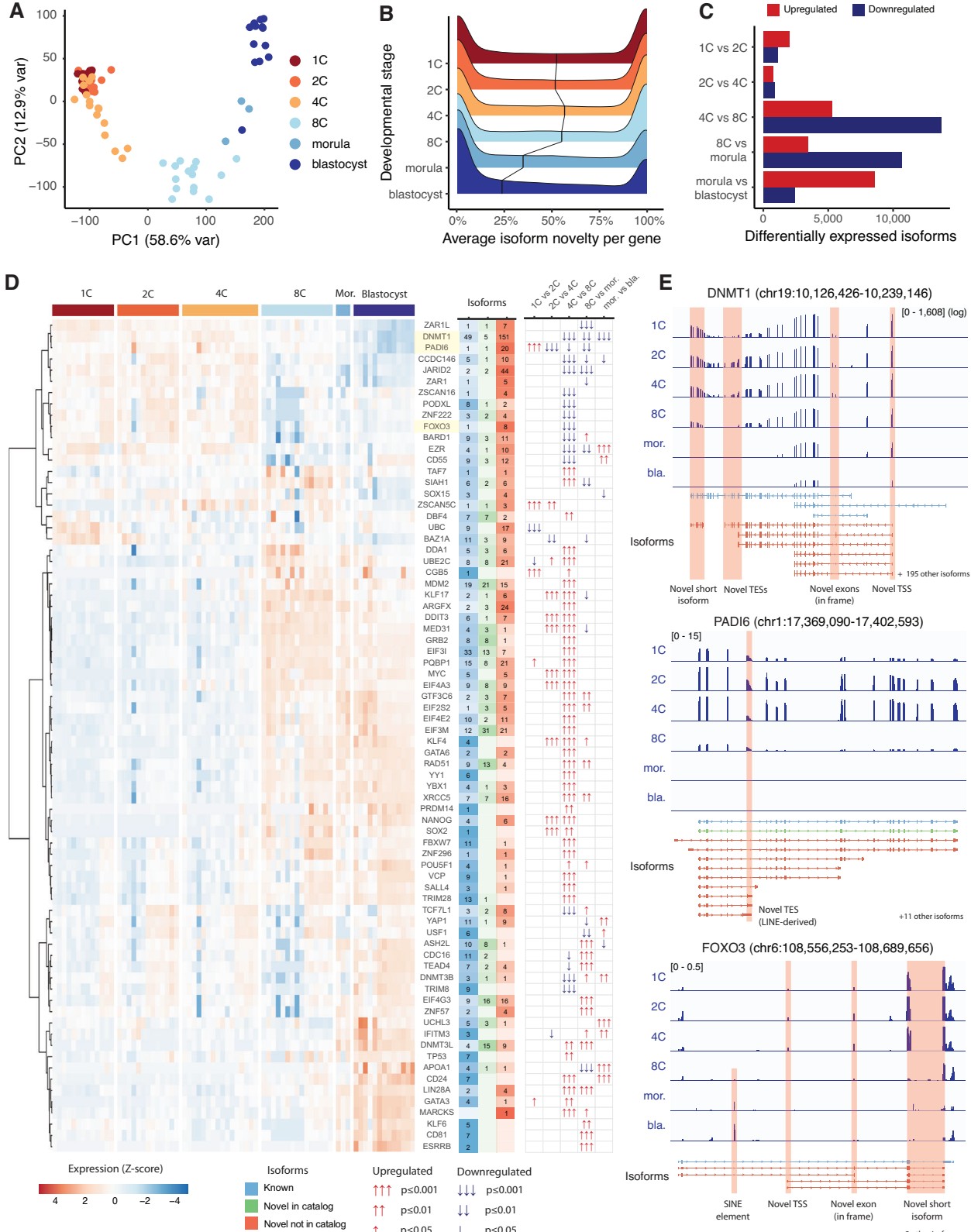

**Fig. 4 | Novel isoform diversity and classifications are associated with known developmental genes. A** PCA of gene expression across short-read RNA-Seq samples, colored by developmental stage. **B** Percentage of short RNA-Seq reads mapping to novel isoforms for each gene expressed in each developmental stage. **C** Number of significantly differentially expressed isoforms at each developmental stage transition. **D** Heatmap of expression levels for selected developmental genes, including the classes of isoforms found for each gene (known, NIC, NNC). The statistical significance and direction of differential gene expression across developmental stage transitions are presented with arrows. **E** Detailed dynamic expression of selected known and novel isoforms of three representative developmental genes: DNMT1, PADI6 and FOXO3. Source data are provided as a Source Data file.

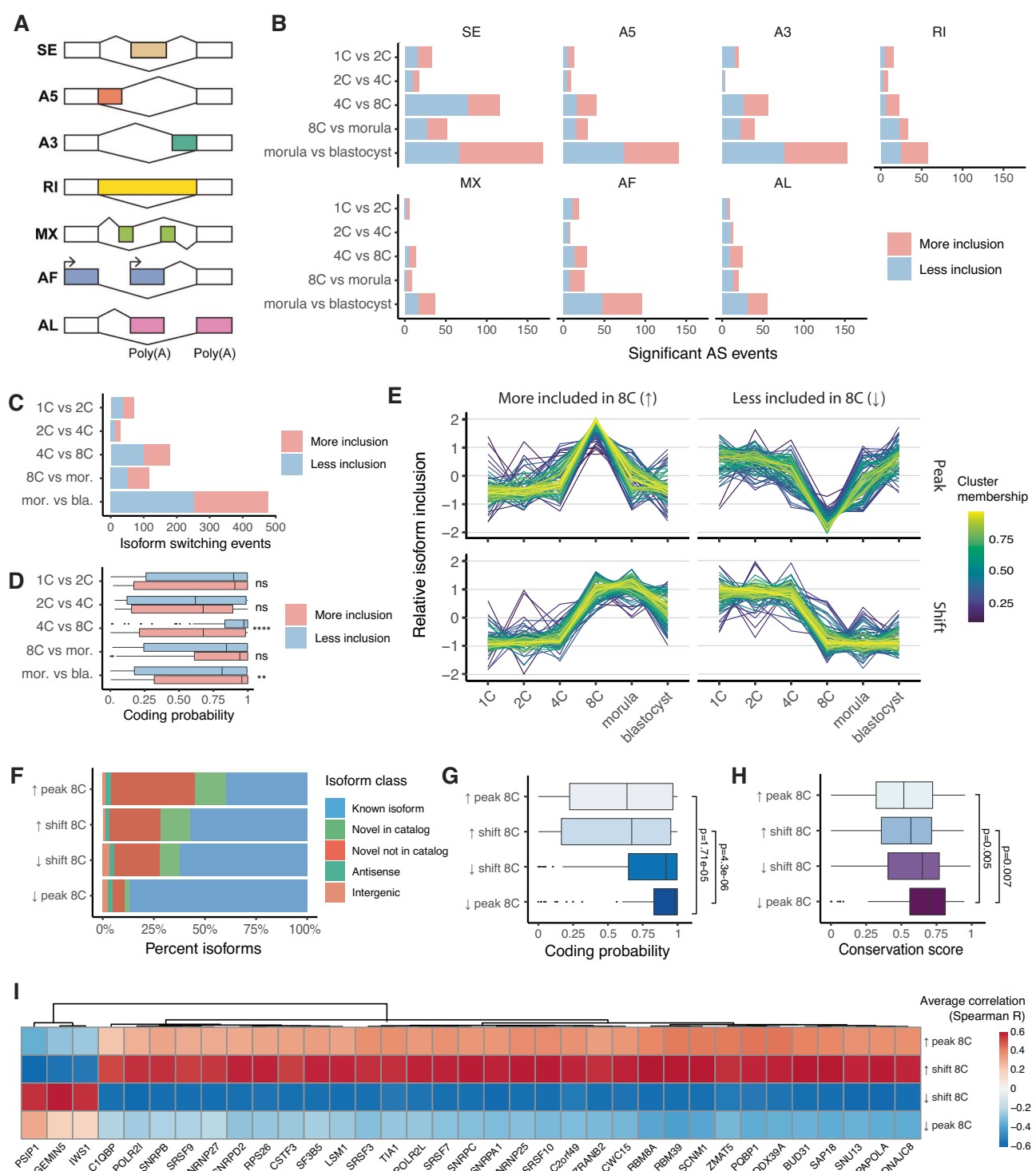

(Fig. 5C). Isoforms that are more highly included at the 8C stage have significantly lower coding probability than isoforms which are less included at this stage, suggesting that AS events lead to ORF disruption during EGA. By contrast, the opposite pattern was observed in the morula-to-blastocyst transition (and in the 8C-to-morula transition, though not statistically significant), suggesting that disrupted ORFs are re-established in subsequent stages (Fig. 5D). To further confirm this, we clustered relative isoform inclusion levels across stages using Mfuzz[91], focusing on isoforms undergoing at least one statistically significant switch over time. We identified four clusters of isoforms undergoing significant switching events at the 8C stage (Fig. 5E, Supplementary Data 8), which are distinguished by the direction of

inclusion (more or less included at the 8C stage) and the inclusion dynamics over time (transient peak or persistent shift). Isoforms with a transient peak of inclusion at the 8C stage are mostly novel and have the lowest coding probability and evolutionary conservation levels among these clusters, while isoforms that are transiently excluded at the 8C stage are predominantly known, and significantly more protein-coding and conserved ($p < 1 \times 10^{-4}$, Wilcoxon rank-sum test, Benjamini-Hochberg correction). By contrast, isoforms undergoing both positive and negative shifts of inclusion display intermediate levels of isoform novelty, coding potential, evolutionary conservation (Fig. 5F–H). Peak 8C-included isoforms have also significantly fewer and shorter introns than their more excluded counterparts (Supplementary Fig. 5B).

**Fig. 5 | Alternative splicing induces ORF disruption and novel isoform inclusion during embryonic genome activation. A** Schematic representation of the seven types of AS events analyzed: skipped exon (SE), alternative 5′ splice site (A5), alternative 3′ splice site (A3), retained intron (RI), mutually exclusive exons (MX), alternative first exon (AF), and alternative last exon (AL). **B** Number of significant AS events at each developmental stage transition. **C** Number of significant isoform switching events at each developmental stage transition. **D** Predicted protein-coding probability of isoforms undergoing significant isoform switches at each developmental stage transition, grouped by direction of inclusion. (1C vs 2C, $p = 0.94$; 2C vs 4C, $p = 0.83$; 4C vs 8C, $p = 4.3e−06$; 8C vs morula, $p = 0.28$; morula vs blastocyst, $p = 0.0021$, unpaired two-sided Wilcoxon Rank Sum test, Benjamini-Hochberg correction; for 1C vs 2C, $n = 35$ more included and $n = 36$ less included isoforms; for 2C vs 4C, $n = 16$ more included and $n = 13$ less included isoforms; for 4C vs 8C, $n = 82$ more included and $n = 99$ less included isoforms; for 8C vs morula, $n = 64$ more included and $n = 52$ less included isoforms; for morula vs blastocyst, $n = 226$ more included and $n = 252$ less included isoforms). **E** Relative inclusion levels of isoforms undergoing significant switching events at the 8C stage. Four major clusters are identified, based on the direction of inclusion (more or less included at the 8C stage) and temporal inclusion dynamics (transient peak or sustained shift). **F** Structural classes of the isoform clusters defined in **E**. **G**, **H** Coding probabilities and evolutionary conservation scores for the isoform clusters defined in **E**. (unpaired two-sided Wilcoxon Rank Sum test, Benjamini-Hochberg correction; cluster sizes: $n = 80$ for positive peak at 8C; $n = 92$ for positive shift at 8C; $n = 93$ for negative shift at 8C; $n = 85$ for negative peak at 8C. Box plot colors are proportional to median values for each corresponding box). **I** Heatmap displaying average correlation (Spearman R) between splicing factor expression and isoform inclusion levels for each isoform cluster defined in **E**. Displayed are the top 35 splicing factors as ranked by highest absolute correlation values. For the box plots in **D**, **G**, **H**, box limits extend from the 25th to 75th percentile, while the middle line represents the median. Whiskers extend to the largest value no further than 1.5 times the inter-quartile range (IQR) from each box hinge. Points beyond the whiskers are outliers. Source data are provided as a Source Data file.

Interestingly, we found over 80 genes transcribing pairs of isoforms belonging to clusters with opposite inclusion patterns throughout development, which may be associated with functional changes not detectable at the gene level (Supplementary Fig. 5C). Together, these results suggest the presence of splicing disruption during EGA, leading to the inclusion of predominantly novel, non-coding isoforms with poorly evolutionarily conserved sequences. This is consistent with recent findings reporting ORF-disrupting exon inclusion during EGA by short-read RNA-Seq analysis[92], as well as studies revealing that splicing inhibition can induce a totipotent, EGA-like state in both mouse and human embryonic stem cells[93,94]. Our analysis is the first to show these patterns at the isoform resolution.

To relate these dynamic changes in AS to splicing factors (SFs), we calculated the correlation between the expression of annotated SFs and relative isoform inclusion levels throughout development, highlighting the top SFs with highest average absolute correlation to the previously identified isoform clusters (Fig. 5I). These include SNRPB and SNRPD2, whose mouse orthologs were recently shown to regulate EGA-associated exon skipping[92]. We then integrated ENCODE eCLIP data for all available SFs to build a network of highly correlated SF-isoform pairs with evidence of SF binding to the isoform nucleotide sequence (Supplementary Fig. 5D), further refining the results. Among the highly correlated pairs we highlight *MRPS21*, a mitochondrial ribosomal gene, whose first intron is bound by the SF TIA1. Both genes are upregulated during EGA, and expression of *TIA1* is significantly correlated to the relative inclusion levels of the first intron of *MRPS21* throughout development (Supplementary Fig. 5E, F). While this analytical approach alone doesn't allow to establish a direct mechanistic link between AS events and SFs, it may be useful to prioritize candidates for further investigation. The results of these analyses are available in Supplementary Data 8.

## Co-expression network analysis of known and novel genes
Next, we investigated the co-expression dynamics of known and novel genes using an unbiased, systems-level approach. To achieve this, we performed weighted gene co-expression network analysis (WGCNA)[95], which identified 30 distinct groups of co-expressed genes (termed modules) whose expression is significantly correlated across samples (Fig. 6A). Novel genes were found to be significantly overrepresented in modules reaching peak expression between the 1C through 8C stages, and significantly underrepresented in modules more highly expressed in the morula and blastocyst ($p < 1.05 \times 10^{-10}$, Fisher's exact test, Benjamini-Hochberg correction, Supplementary Fig. 5G), further confirming that they are most highly expressed in earlier pre-implantation stages and downregulated in the morula and blastocyst.

We further investigated 6 selected modules which recapitulate stage-specific developmental patterns and display significantly enriched gene ontology terms, as well as diverse patterns of novelty

(Fig. 6B–D). Modules 1 and 5, whose genes are broadly expressed between the 1C to 4C stages and are subsequently downregulated, are composed of 23% and 14.5% novel genes respectively. Collectively, these modules comprise over 4000 genes involved in cell signaling, adhesion and cytoskeletal organization, including key regulators such as *KMT2C*, *MDM4*, *DNMT1*, *FOXO3* and *PADI6*. Many of these genes likely include maternally inherited mRNAs, whose expression, splicing and translational efficiency is typically additionally regulated by cytoplasmic polyadenylation prior to EGA in mammals and other organisms[96–99]. Module 30, whose genes display a transient peak of expression at the 4C stage, was the most novel of the highlighted clusters. Over half of this module is comprised of previously unknown antisense and intergenic genes, while its known genes are involved in pathways including cell signaling and transduction. Module 23, whose genes are transiently upregulated at the 8C stage, is similarly comprised by over 40% novel genes, and includes known regulators of transcription, cell proliferation and apoptosis such as *YAP1* and *SMARCD1*. By contrast, modules associated with later developmental time points are almost entirely comprised by known genes. Module 3, which is activated at the 8C stage and contains known pluripotency markers such as *SOX2*, *NANOG*, *KLF4*, and other genes involved in RNA processing and translation, contains only ~1% novel genes. Module 7, whose genes are activated in the blastocyst and include *GATA3*, *KLF6* and *ESRRB*, contains a similarly small number of novel genes.

We further assessed the validity of these modules by performing a preservation analysis using RNA-Seq data from published human, macaque and marmoset preimplantation embryo studies (Fig. 6E). All six highlighted modules were significantly conserved across these orthogonal datasets ($p \leq 0.0001$, Bonferroni correction), with the only exception of module 30 in the marmoset dataset. The five most conserved modules likely represent well-established, evolutionary conserved networks of genes which play key roles in preimplantation development across species. Module 30 may instead represent more recent evolutionary developmental programs specific to macaque and human. Indeed, only 25.2% of the genes in this module are fully mapped and expressed at corresponding genomic coordinates in marmoset embryos, compared to 47.6% in the macaque (Supplementary Fig. 5H). In addition, the coding potential analysis revealed that modules 30 and 23 are predominantly comprised of predicted non-coding genes (74% and 80.4% respectively), with the other modules displaying a larger proportion of predicted protein-coding genes (Supplementary Fig. 5I). We further scanned the 3′ UTR sequences of genes in each module for miRNA binding sites using miRanda[100], identifying miRNAs that are significantly predicted to bind module 1 and 5 genes by overrepresentation analysis (Supplementary Fig. 5J–K). The global gene network, alongside a selection of key developmental genes in highlighted modules, are displayed in Fig. 6F.

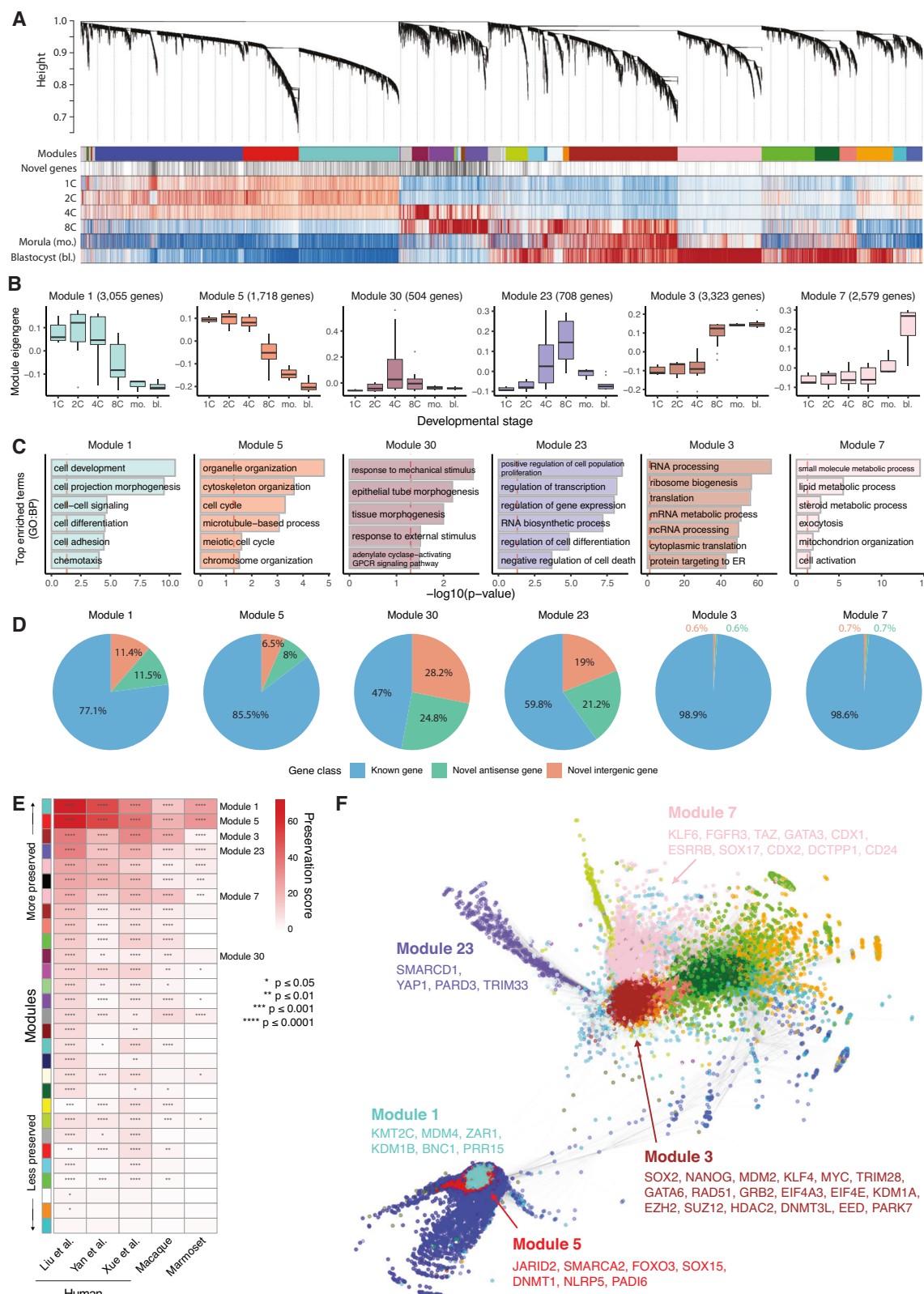

Consistent with previous findings by Xue et al. [23], these results demonstrate that human preimplantation transcriptome dynamics may be recapitulated by key modules of functionally defined co-expressed genes. Crucially, by leveraging isoform-resolved data and a larger sample size, we were able to identify additional undiscovered gene modules, including two 4C and 8C-specific modules composed of hundreds of novel genes.

## Investigating unannotated genes in early human embryos

We further characterized the biological properties and transcriptional regulators of the thousands of unannotated genes by integrating multiple predictive tools. First we performed soft clustering using Mfuzz[91], identifying five clusters of novel genes with distinct embryonic stage-specific expression (Fig. 7A). Most novel genes were assigned to two clusters with peak expression at either the 1C or the 2C phases

**Fig. 6 | Novel genes are key components of early expressed, evolutionarily conserved modules of co-expressed developmental genes. A** Hierarchical clustering tree displaying results of the gene co-expression network analysis. Modules of genes co-expressed across developmental stages are displayed as color bars. Novel genes are highlighted below. Normalized gene expression across stages is also displayed (red - highest relative expression, blue - lowest expression). **B** Representative expression profiles (module eigengenes) of selected gene modules characterizing specific developmental stages (sample sizes for each stage shown in Fig. 1A). **C** Gene Ontology: Biological Process terms enriched in the selected gene modules shown in **B**. **D** Percentage of each gene class in the selected modules. **E** Heatmap of module preservation scores in independent, publicly

available short-read RNA-Seq datasets of human and non-human primate embryos (scores and *p*-values calculated using the WGCNA modulePreservation function, *p*-values adjusted using Bonferroni method). Module colors (from **A**) are shown on the left. **F** Network diagram displaying connected genes from different modules (represented as different colors, from **A**) of the gene co-expression network. A selection of known developmental genes is highlighted for five modules. For the box plot in **B**, box limits extend from the 25th to 75th percentile, while the middle line represents the median. Whiskers extend to the largest value no further than 1.5 times the inter-quartile range (IQR) from each box hinge. Points beyond the whiskers are outliers. Source data are provided as a Source Data file.

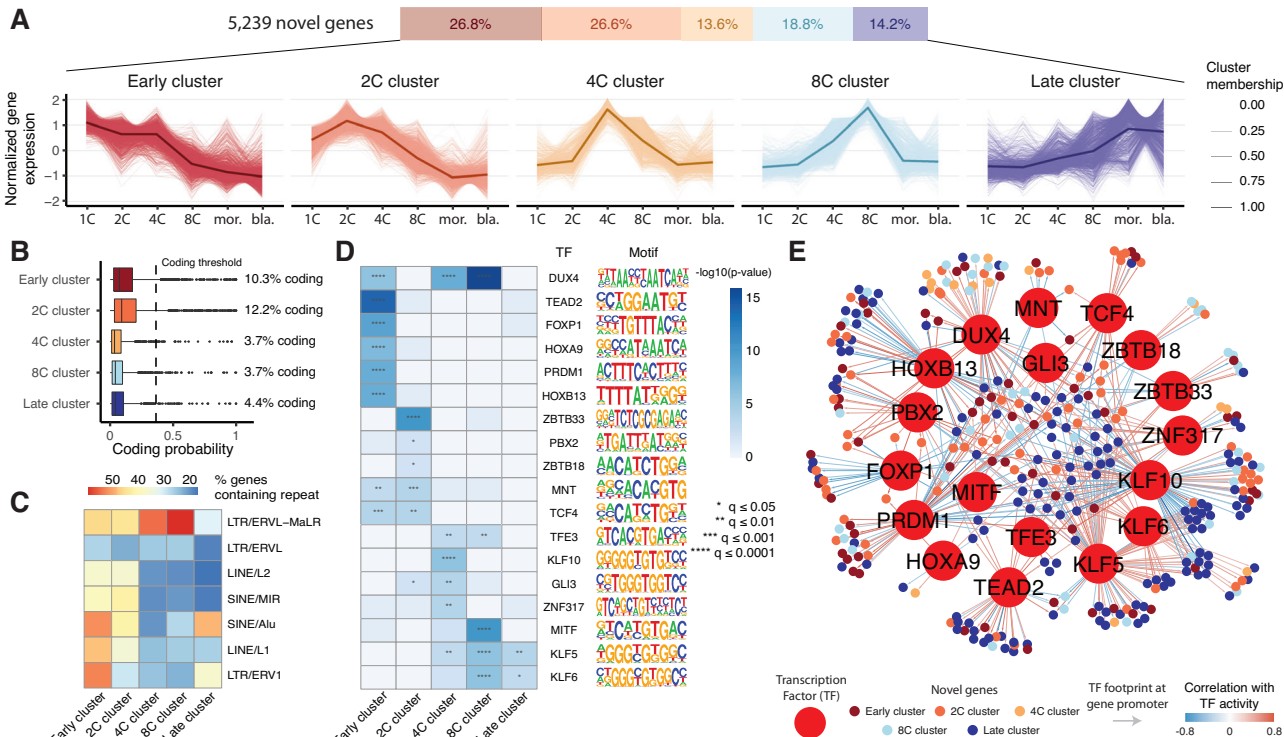

**Fig. 7 | Novel genes have distinct expression patterns, predicted biological properties, and transcriptional regulators. A** Novel genes separate into five clusters with distinct expression patterns across human preimplantation development (*n* = 1402 genes in early cluster, 1391 genes in 2C cluster, 712 genes in 4C cluster, 987 genes in 8C cluster, 746 genes in late cluster). **B** Predicted coding probability for novel genes in each cluster defined in **A**. **C** Heatmap displaying the percentage of novel genes in each cluster containing distinct classes of human retrotransposons. **D** Heatmap displaying predicted transcription factor regulators of each novel gene cluster, including their DNA binding motifs (*p*-values calculated

using HOMER findMotifsGenome.pl, *q*-values adjusted using Benjamini-Hochberg method). **E** Network diagram displaying predicted TFs-novel target gene pairs, as determined by integration of ATAC-Seq footprinting and inferred TF activity-gene expression correlation. For the box plot in **B**, box limits extend from the 25th to 75th percentile, while the middle line represents the median. Whiskers extend to the largest value no further than 1.5 times the inter-quartile range (IQR) from each box hinge. Points beyond the whiskers are outliers. Source data are provided as a Source Data file.

(termed early and 2C clusters respectively, comprising 53% of novel genes in total). Another two clusters of genes are transiently upregulated during the 4C and 8C stages (14% and 19% genes respectively), and the late cluster is primarily activated in either the morula or blastocyst (14% of genes). The expression patterns of these clusters are supported by data from previously published human embryo short-read RNA-Seq studies (Supplementary Fig. 6A). While only a small fraction of these genes is fully mapped and detected at corresponding genomic coordinates in the macaque and marmoset preimplantation embryos, their expression over time is broadly consistent with the human, further supporting their validity and suggesting a conserved role in primate embryo development (Supplementary Fig. 6B). Early and 2C gene clusters, unlike others, are also broadly detected in oocytes (Supplementary Fig. 6C), indicating they likely represent maternally inherited genes. Reanalysis of publicly available scRNA-Seq data revealed that these gene clusters are broadly expressed in E3-

stage embryos, but not in 8CLCs (Supplementary Fig. 6D). Gene clusters also display significant differences in their predicted biological properties. While the majority of such genes are predicted to be non-coding, early expressed genes are predicted to be more protein-coding (>10%) than their later-expressed counterparts (~4%), suggesting the presence of hundreds of maternally-inherited or early activated protein-coding genes which are yet uncharacterized (Fig. 7B). TE content was also found to vary across gene clusters (Fig. 7C). Early expressed genes are associated with a wide variety of repetitive elements, including multiple classes of LTRs, LINEs and SINEs. Interestingly, over 70% of 4C and 8C cluster genes contain LTR/ERVL-MaLR family repeats, including HERVH-int, MLT2A1 and MLT2A2, all of which have been previously reported to be highly activated during the 8C embryonic phase, but never shown to form chimeric transcripts using isoform-resolved data. Late-expressed genes display the lowest levels of repetitive element integration, primarily consisting of SINE/Alu and

LTR/ERV1 elements. We further predicted novel gene function by integrating annotations of the most strongly co-expressed known genes, an approach that has been used to infer putative roles of unannotated loci in a variety of contexts[101–103]. Early-expressed novel genes were found to be co-regulated with known genes involved in cell signaling, adhesion, and cellular component morphogenesis, while later expressed ones are instead co-expressed with genes involved in DNA-templated transcription, mRNA processing and splicing (Supplementary Fig. 6E).

We next predicted transcriptional regulators of novel gene clusters by performing a motif analysis using HOMER[104] (Fig. 7D). Promoter regions of early expressed genes were enriched for binding sites of TFs including DUX4, TEAD2, and FOXP1, which are known to play key roles in the regulation of EGA, stem cell self-renewal and differentiation[30, 105–107]. Interestingly, DUX4 was also strongly predicted to bind promoters of 4C and 8C cluster genes. This is consistent with the high abundance of ERVL/MaLR repeats among such loci, which have been previously shown to be bound by DUX4[108]. Late expressed gene promoters were instead predicted to be bound by blastocyst fate TFs, including kruppel-like factors such as KLF5[109]. To further examine regulatory interactions between TFs and novel genes, we performed an ATAC-Seq footprinting analysis using TOBIAS[110], which predicted TFs bound at novel gene promoters by integration of chromatin accessibility data from Liu et al. with information on known TF binding motifs. Predicted TF-gene pairs were further refined by requiring a statistically significant correlation between the target gene expression and the TF activity as inferred by VIPER[111]. We thus built a filtered TF-novel gene interaction network (Fig. 7E). Consistently with the motif analysis, MNT was predicted to predominantly regulate 2C and 4C clusters, DUX4 to regulate mainly 4C and 8C-cluster genes, while genes such as KLF5, KLF6 and TEAD2 instead predicted to mostly regulate late cluster genes. Together, these predictions shed light on the putative biological function and transcriptional regulators of the thousands of newly identified genes, and will empower future studies seeking to further understand their function in early human development.

We next showcase selected examples of novel genes with diverse patterns of evolutionary conservation, predicted function, and expression dynamics throughout development. First, we highlight *NOVELG000067783*, a novel, early-expressed protein-coding gene located on human chromosome 3 which does not overlap any known annotations (Fig. 8A). The locus transcribes multiple isoforms, including several alternative splice variants predicted to encode peptides containing ferritin domains, as well as non-coding RNAs (Supplementary Fig. 7A). In addition to being supported across all integrated human embryo short-read RNA-Seq datasets (Supplementary Fig. 7B), the gene is also detected at syntenic genomic locations in both macaque and marmoset preimplantation embryos, displaying similar expression patterns throughout development (Fig. 8B, C). Interestingly, the gene is more lowly expressed and lacks its first exon in the marmoset, suggesting it may be a product of more recent evolutionary events. We also highlight three examples of novel predicted lncRNAs that are expressed in 1-4 cell embryos and are supported by all integrated human embryo short-read RNA-Seq and ATAC-Seq datasets, but not detected in either of the primate embryo studies, thus representing human-specific transcriptional events (Fig. 8D, Supplementary Fig. 7C). These include *NOVELG000084291*, a novel antisense gene which shares its TSS with known protein-coding gene UNC13B, but is transcribed in the opposite strand; *NOVELG000070644*, a novel intergenic gene located on chromosome 16; and *NOVELG000059671*, a novel antisense gene which overlaps inferred pseudogene *PPIAP24* on the opposite strand. *NOVELG000084291* displays abundant integration of HERVH-int and LTR7, and further contains a SINE/Alu repeat within its first intron. Its expression is anticorrelated to both its antisense neighbor *UNC13B* (Supplementary Fig. 7D) and its intronic SINE,

suggesting that it may act as a natural antisense transcript[112] and pointing to a potential TE-mediated role in gene expression regulation[113]. The TSSs of most *NOVELG000070644* and *NOVELG000059671* isoforms originate from insertions of THE1D, an LTR element of the ERVL-MaLR family, which are activated by DUX4 and have been shown to provide alternative promoters for multiple genes in placenta and lymphoma studies[114–116]. While these insertions are present in both macaque and marmoset genomes, neither of the genes are detected in the corresponding primate embryos. The expression of both genes is significantly correlated to VIPER-inferred DUX4 activity across preimplantation stages (Supplementary Fig. 7E), and *NOVELG000059671* further displays predicted DUX4 binding footprints at its TSS at the 4C and 8C stages, thus suggesting its expression may be regulated by this TF. Notably, we further validated the expression of novel genes by PCR on an additional set of embryos (Supplementary Fig. 8), as well as in multiple independently published short-read RNA-Seq datasets from human preimplantation embryos (Supplementary Fig. 9). Together, these results show that the novel transcriptome contains both evolutionarily conserved and human-specific novel genes and isoforms, with a wide variety of predicted biological properties and transcriptional regulators. These examples only showcase a small fraction of the novel genes and isoforms described in this study, and we anticipate that this data will serve as a valuable resource to empower future studies seeking to further understand early development.

## Discussion

Here, we present the first isoform-resolved human preimplantation reference transcriptome generated by combined long- and short-read RNA-Seq, in silico validated by integrating existing embryo multi-omics datasets, and extensively characterized to predict the biological relevance of thousands of unannotated genes and isoforms. Using this comprehensive computational approach, we identified 30,988 unannotated isoforms transcribed from known gene loci consisting of a novel combination of known splice sites, and 79,224 unannotated isoforms transcribed from known loci containing at least one novel splice site. We also identified 17,964 isoforms transcribed from 5239 previously uncharacterized loci which overlap known genes on the opposite strand, or are located in intergenic space. The full set of isoforms, associated predictions and integrated datasets can be freely accessed at the following online resource: https://denis-torre.github.io/embryo-transcriptome/.

Integration of multiple computational predictive tools and analytical approaches allowed us to gain insights into the human preimplantation embryo transcriptome at unprecedented resolution. Thorough characterization of the 5239 newly identified genes revealed that these are largely predicted to be non-coding, rich in TEs and poorly evolutionarily conserved beyond hominids, underscoring that common models to investigate mammalian development such as the mouse may not fully recapitulate many of these early transcriptional events in humans. Indeed, it is known that TEs can contribute to the generation of novel lncRNAs[43, 117], including human endogenous retrovirus (HERV)-K and HERV-H elements[118]. Our catalog greatly expands the number of TE-chimeric isoforms, with thousands of such unannotated transcriptional events detected in human preimplantation stages. Prior to the release of this transcriptome, the repetitive nature of such sequences would have made the reconstruction of these isoforms difficult using existing lower resolution datasets. Most of these genes are either maternally inherited or transiently expressed during EGA, as underscored by analysis of our standalone data and the orthogonal integrated published short-read RNA-Seq data. Further analysis of these isoforms will increase our understanding in the pervasive role of TE-chimeric promoters in preimplantation development, though it remains to be determined to what extend these new gene modules comprise key mechanistic players of human preimplantation

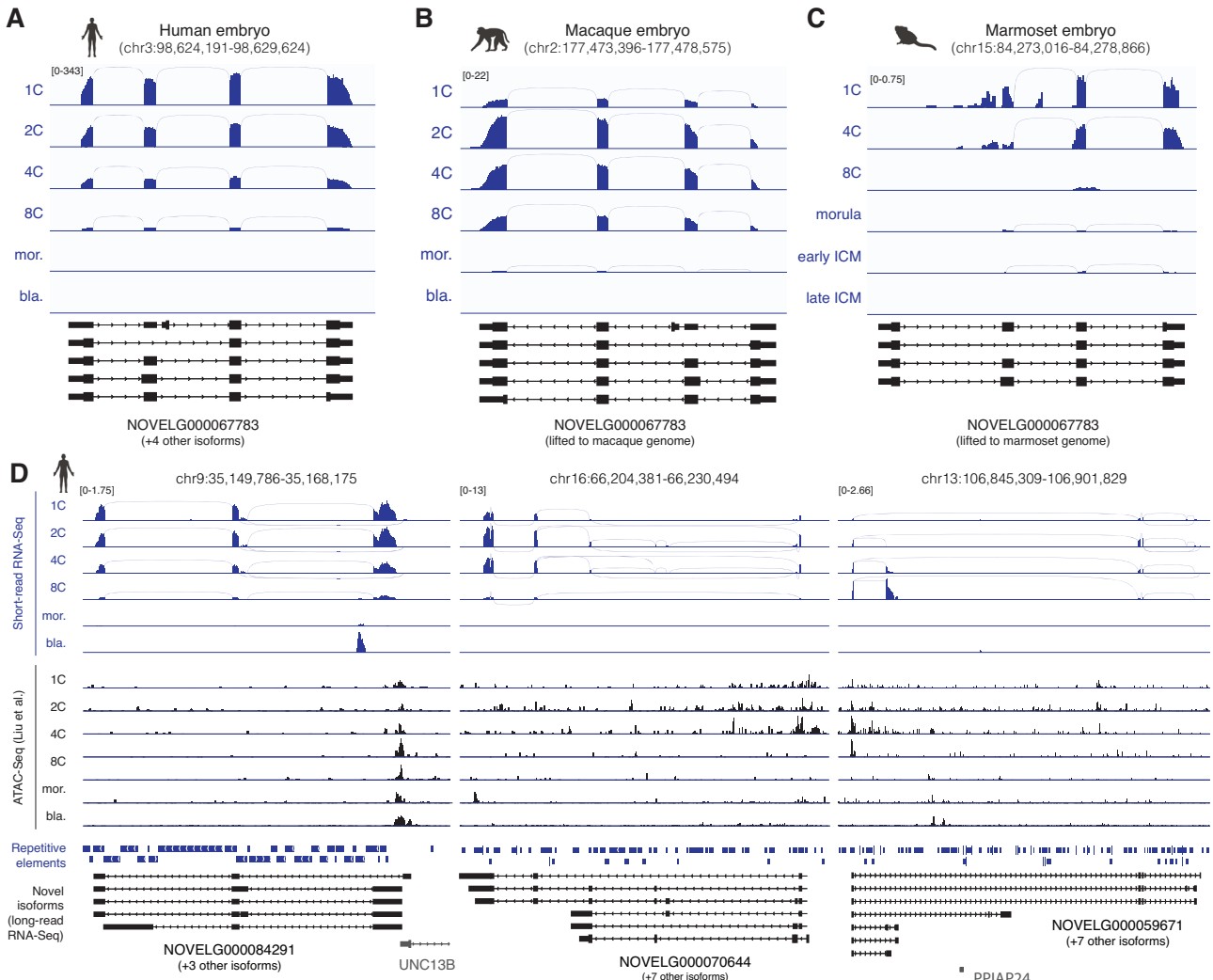

**Fig. 8 | Examples of novel genes expressed in human and non-human primate embryos. A** Short-read RNA-Seq expression across developmental stages (displayed in blue, RPKM normalization) and long-read-defined isoforms for evolutionarily conserved novel human gene *NOVELG000067783*. **B, C** Short-read RNA-Seq expression from macaque and marmoset preimplantation embryos (data from Wang et al. and Boroviak et al., respectively), and long-read-defined isoforms for novel human gene *NOVELG000067783*. Novel isoform annotations from the human genome (hg38) were lifted to the corresponding locations in the respective primate genomes (Mmul10 for macaque, calJac4 for marmoset) using liftOver. **D** Short-read RNA-Seq expression, long-read-defined isoforms and matching chromatin accessibility (from Liu et al.) for three novel human-specific genes. Also shown are repetitive genomic elements from RepeatMasker. Source data are provided as a Source Data file. Icons in **A**–**D** were created with BioRender.com.

development, or transcriptional by-products of this highly dynamic and complex process.

In addition to novel genes, we found widespread evidence of unannotated alternative splicing events taking place in known genes, including known regulators of early development such as *DNMT1*, *FOXO3*, and *PADI6*, further underscoring the necessity of leveraging long-read RNA-Seq for improving annotations for transcriptomic analysis, especially from relatively understudied conditions such as human preimplantation development. Further functional work will be able to provide specific answers on the function of individual isoforms for known genes. Nonetheless, our analysis was also able to identify global patterns, specifically taking place during EGA, which exhibits transient inclusion of unannotated, poorly evolutionarily conserved isoforms, as recently reported in a study leveraging short-read RNA-Seq[92].

Our study builds upon the results of previous publications investigating human preimplantation development using multi-omics approaches such as bulk and single-cell transcriptomics[21–25,27], analysis of chromatin accessibility[22,119,120], histone modifications[54] and DNA

methylation[58,121]. While these studies greatly increased our understanding of the transcriptomic and epigenomic events taking place in these early stages, they relied on the limited, largely short read-derived transcriptome annotations available at the time, which fail to capture the full length of most mRNAs. Integration and reanalysis of data from these studies revealed that the novel genes and isoforms described in this work are widely supported at both the transcriptional and epigenetic levels. Many novel isoforms are also supported by transcriptomic data generated from macaque and marmoset preimplantation embryos[65,66], suggesting that some of these unannotated events are also present in other primates. Multi-omics studies on mouse embryos have also been conducted[120,122–125], including a recent study using both long- and short-read RNA-Seq, which identified 6289 novel isoforms from previously annotated genes and 2280 from unannotated genes[15], though at a lower sequencing depth and smaller sample size compared to the dataset presented here. Recently, an orthogonal study was published using long read RNA-Seq to characterize the poly(A) tail length during the maternal-to-zygote transition of human preimplantation embryos, rather than alternative splicing[126]. However, our

dataset is the first study presenting an isoform-resolved transcriptome conducted on human embryos spanning the zygotic to blastocyst stages of human preimplantation development.

Experimental investigation of early human embryo development remains challenging using real-time embryo manipulation in a laboratory setting. However, recent studies have described novel experimental platforms to study early developmental stages starting from human pluripotent stem cells (hPSCs): 8C-like cells (8CLCs), which display EGA-like transcriptional and epigenetic features[52,94]; and blastoids, blastocyst-like structures that develop the three lineages (trophectoderm, epiblast, primitive endoderm) typical of normal human development[53,127,128]. The integration of single-cell RNA-Seq data these models confirmed widespread expression of the novel isoforms and genes in 8CLCs and blastoids, confirming that these platforms recapitulate unannotated transcriptional events taking place in early human embryos, and indicating that they may be leveraged to further understand their function.

Together, our results greatly expand the annotation of isoform diversity in human preimplantation development, revealing tens of thousands of unannotated isoforms transcribed from both known and novel genes. Integration of diverse computational tools and multi-omics datasets further validates these isoforms and helps predict their putative biological function. By providing these results and interactive database to the community, we anticipate that our work will help guide future experimental studies aiming to explore the role of critical genes in development and disease.

## Methods

### Ethics statement
Embryos were produced by IVF for clinical purposes between years 1997 and 2017 at Tel Aviv Medical Center and surplus embryos at different preimplantation development stages (from zygote to blastocyst stage, see Supplementary Data 1) were cryopreserved for future use. The embryos used in this study were spare frozen preimplantation human IVF embryos at day 1-6 of development (after fertilization), that were donated by IVF patients after they have completed family planning, and after signing a full informed consent. The informed consent was used in compliance with Institutional Review Board following approval by the National Ethics Committee (IRB 559/16: "Advanced RNA sequencing technologies for characterization of human preimplantation embryo's transcriptome")". Only embryos that were donated for research were allocated for this study. These human embryos represent the infertile population. Embryos were thawed according to their day of freezing (i.e. developmental designation), and according to the study design aimed to extract RNA from the embryo cohort used for our study across each preimplantation stage.

### Embryo selection
Most embryos analyzed (76%) were at high/good quality when frozen as well as at thawing for RNA preparation. Embryos were scored according to the Istanbul consensus workshop on embryo assessment: proceedings of an expert meeting[129]. The spare IVF embryos used in this study are considered genetically normal, as the biological parents performed IVF due to infertility problems but otherwise have no indications of genetic abnormalities. However, we cannot rule out the possibility that some of them may carry genetic mutations that were not known or diagnosed at the time the embryos were frozen. Furthermore, it is well accepted that some IVF embryos may be aneuploid[130].

### cDNA preparation
Donated embryos were thawed using the Quinn's Advantage Thaw Kit (SAGE) following the manufacturer's instructions. Single embryos were lysed to release mRNA, which was primed using a modified oligo-dT primer and reverse-transcribed using template switching technology to generate full-length cDNA using the SMART-Seq v4 Ultra Low Input RNA kit (Clontech, Takara; Cat# 634897), following the manufacturer's instructions. The first-strand cDNA templates were then amplified using 14-16 cycles of LD-PCR and then purified using AMPure XP beads (Beckman Coulter). The resulting double-stranded cDNA templates were then transferred to the Genomics Technology Facility at the Icahn School of Medicine at Mount Sinai for sequencing. Each sample was aliquoted for both PacBio and Illumina library preparation.

### SMRT-seq library construction and sequencing
Full-length cDNA was used as input for preparing SMRTbell libraries using the SMRTbell Express Template Preparation Kit v2.0 as recommended by the manufacturer (Pacific Biosciences). Samples with enough cDNA mass (>100 ng) were prepped as individual libraries and those with little mass available (<100 ng) were pooled together for library prep. Briefly, the cDNA was treated with a DNA Damage Repair mix to repair nicked DNA, followed by an End Repair and A-tailing reaction to repair blunt ends and adenylate each template. Next, overhang SMRTbell adapters are ligated onto the ends of each template and purified with 0.6X AMPure PB beads to remove small fragments and excess reagents (Pacific Biosciences). The completed SMRTbell libraries were then annealed to sequencing primer v4 and bound to sequencing polymerase 2.0 before being sequenced using one SMRTcell 8 M on the Sequel II system with a 24-hour movie.

### Illumina RNA-seq library construction and sequencing
Illumina sequencing libraries were prepared by following the Nextera XT DNA Library Preparation Kit (Illumina, # FC-131-1024) workflow, as recommended by the manufacturer. Briefly, 0.7–1 ng of amplified full-length cDNA from each sample was ultrasonically sheared using a Covaris AFA system, while also simultaneously ligated with adapters by tagmentation. Individual indices were then ligated onto the tagmented cDNA templates via PCR. The libraries were sequenced on an S1 200 flowcell on the NovaSeq platform as 100 nt or 125 nt paired-end reads at a depth of 50 million reads per sample.

### PacBio long-read sequencing primary data processing
The developer version of the PacBio Iso-Seq3 pipeline (v3.4.0) was used for preparing full-length non-concatemer (FLNC) reads from the raw sequencing data. First, subreads were intramolecular error-corrected and polished using the circular consensus sequencing (CCS) algorithm (v5.0.0) to produce highly accurate (>Q10) CCS reads, each requiring a minimum of 1 complete polymerase pass. The polished CCS reads were then passed to the lima tool (v2.0.0) to remove barcodes (if used), SMART-Seq primers and template-switching oligo sequences and orient the isoforms into the correct 5' to 3' direction. The refine command was then used to remove polyA tails and concatemers to generate FLNC reads ready for downstream analysis. The FLNC reads per Sequel II SMRTcell were then mapped to the GRCh38 genome assembly using the splice-aware aligner, minimap2 (v2.17)[131], with the following parameters: minimap2 -ax splice -uf --secondary=no -C5 --MD. Unmapped full-length reads, and reads with <50 MAPQ, were removed using sambamba (v0.5.6)[132].

### Short-read RNA-seq primary data processing
Short-read RNA-Seq raw reads underwent adapter trimming and quality control using Trim Galore (v0.6.6)[133] with default parameters. Filtered reads were initially aligned to the GRCh38 reference genome together with the Ensembl v102 gene annotation reference database with STAR (v2.7.5b)[134] using the two-pass mapping approach in order to count reads spanning splice junctions. Spliced read count was provided to filter long reads (see below).

## Integration of short and long RNA-Seq reads to build the isoform-resolved embryonic transcriptome

Uniquely mapped long- and short RNA-Seq reads were integrated to generate a novel isoform-resolved transcriptome using TALON (v5.0)[34]. First, SAM files containing aligned PacBio FLNC reads were processed using TranscriptClean (v2.0.2)[135] in order to correct mismatches, indels, and to remove full-length isoforms containing non-canonical splice junctions not supported by short reads. Next, talon_label_reads was used to flag FLNC isoforms with evidence of intra-priming artifacts (priming of genomic A-rich tracts during reverse transcription) using default parameters. talon_initialize_database was then used to generate an SQLite database against which to classify PacBio isoforms by providing the Ensembl GRCh38 v102 transcript reference using the following parameters: --l 200 –50 1000 –3p 1000. These parameters require FLNC isoforms to be at least 200 bp long, and instruct TALON to collapse transcript models with congruent internal exons whose 5′ and 3′ ends vary by up to 1000 bp. Next, the talon command was used to collapse, count and classify FLNC isoforms into a reference transcriptome by requiring a minimum alignment threshold of 99% coverage and a minimum sequence identity of 95%. Isoforms were additionally filtered by removing transcripts with a fraction of A > 0.6 (internal priming artifacts). The reference GTF was generated using the talon_create_GTF command, and used to calculate splice junction reads from short-read RNA-Seq data using the STAR two-pass mapping approach with default parameters. Isoforms were subsequently filtered by requiring at least one uniquely mapped spliced short read overlapping each of their junctions in at least three independent short-read RNA-Seq samples. Lastly, SQANTI3 (v4.2)[136] was used to classify isoforms in the reference GTF with default parameters and the provided human polyA motif list. Final isoform classifications were generated by integrating the complementary TALON and SQANTI3 classifications as described in Supplementary Fig. 1F.

## Gene and isoform expression quantification from bulk and single-cell short-read RNA-Seq

Expression of isoforms and genes in the novel reference transcriptome was quantified using our own short-read RNA-Seq samples and publicly available data from the following studies: Liu et al. [22]. (NCBI SRA accession SRP163205), Yan et al. [21]. (SRP011546), Xue et al. [23]. (SRP018525), Petropoulos et al. [24]. (ArrayExpress accession E-MTAB-3929), Mazid et al. [52]. (CNGDB Nucleotide Sequence Archive accession CNP0001454), Kagawa et al. [53]. (SRP323840), Mazin et al. [68]. (E-MTAB-6814). First, short RNA-Seq reads were trimmed from adapters and filtered using Trim Galore as described above. Next, the filtered reads were aligned to the GRCh38 reference genome using STAR (v2.7.5b)[134] two-pass mapping using default parameters. Finally, RSEM (v1.3.3)[137] was used to calculate expression using the following parameters: rsem-calculate-expression –alignments --strandedness none --paired-end --estimate-rspd. Gene expression of large single-cell RNA-Seq datasets generated by SmartSeq (Petropoulos et al., Mazid et al., Kagawa et al.) was quantified using STARsolo (v2.7.9a) with the following parameters: --soloType SmartSeq --soloUMIdedup Exact NoDedup --soloStrand Unstranded –soloMultiMappers EM. Gene counts were imported in R using Seurat (v4.0.0)[138], outlier cells were filtered as described in the respective studies, data was normalized and used to find variable features. To generate the UMAP in Fig. 2D, Petropoulos et al. and Mazid et al. datasets were integrated using the FindIntegrationAnchors and IntegrateData functions from Seurat, followed by data scaling, PCA and UMAP with 30 principal components. Gene set scores for novel antisense and intergenic genes (Fig. 2E) were calculated using the AddModuleScore function on the unintegrated data. Panels for Kagawa et al. (Supplementary Fig. 3A, B) were generated as above, but without data integration. BigWig files displaying short-read RNA-Seq pileup were generated using bamCoverage from deepTools (v3.5.0), by providing scaling factors calculated for each sample using DESeq2

(v1.3.0) using the --scaleFactor parameter. Average coverage for each developmental stage was calculated by averaging the signal across scaled replicates for each developmental stage using the mean function from wiggletools (v1.2).

## Analysis of protein-coding potential, protein domain content and repeat content

Open reading frames (ORFs) were predicted from the nucleotide sequence of each isoform using CPAT (v3.0.2)[36] with default parameters. As recommended by the authors, ORFs with a coding probability ≥0.364 were labeled as protein-coding, while sequences below this threshold were classified as non-coding. For every coding isoform, the best ORF nucleotide sequence was translated into the corresponding amino acid sequence using the translate function from Biostrings (v2.58) in an R 4.0.3 environment, and then scanned for protein domains using pfam_scan.pl (v1.6) and HMMer (v3.3)[139] with default parameters. Isoform nucleotide sequences were also assessed for the presence of repetitive elements by using RepeatMasker (v4.1.1)[42] with default parameters. Genomic annotation of isoform TSS locations compared to known transcripts was performed using ChIPSeeker (v1.26.0)[140].

## Evolutionary conservation analysis

Evolutionary conservation scores were obtained by downloading the hg38.phastcons100way.bw files from the UCSC genome browser database, which contain base-wise conservation scores estimated using PhastCons[49] from multiple alignments of 99 vertebrate genomes to the human genome. Conservation scores across isoforms were calculated using the computeMatrix function from deepTools (v3.5.0) with the following parameters: scale-regions --metagene --beforeRegionStartLength 3000 --regionBodyLength 5000 --afterRegionStartLength 3000. Density profiles were plotted using the plotHeatmap function. For each isoform structural category, a control set of background regions was calculated by randomly shuffling isoforms in intergenic space using the bedtools shuffle function from bedtools (v.2.29.2) by providing transcripts from Ensembl GRCh38 v102 and chromosome gaps to the -excl parameter. Average conservation scores for each isoform and the matching shuffled intergenic regions were calculated using bigWigAverageOverBed (v2) function from UCSC. P-values were calculated by comparing average conservation scores of isoforms to shuffled intergenic regions using the Wilcoxon rank-sum test in an R 4.0.3 environment and adjusted using the Benjamini-Hochberg method. BLAST v2.9.0[51] was used to scan the nucleotide sequences of isoforms from the reference transcriptome against a nucleotide database built using the latest available genome sequence assemblies of 50 selected vertebrates downloaded from UCSC[141]. BLAST was run using -task megablast and default parameters. Isoforms were considered a hit to each target genome if BLAST returned at least one alignment of >100 bp with >95% identity and an E-value < 0.05. The temporal estimates of evolutionary divergence from hominids displayed in the phylogenetic tree in Fig. 1I were calculated using TimeTree[142,143].

## Analysis of publicly available ATAC-seq and CUT&RUN data

Publicly available human embryo ATAC-Seq data was downloaded from the SRA database (Liu et al. [22], SRP163205), while human embryo CUT&RUN data was downloaded from the European Nucleotide Archive (Xia et al. [54], PRJNA513257). Raw ATAC-Seq reads underwent adapter trimming and quality control with Trim Galore (v0.6.6) using default parameters for paired-end data. Since the CUT&RUN dataset contained both single- and paired-end samples with variable read lengths between 50 bp and 150 bp, additional steps were taken to ensure that the differences in sequencing configuration would not introduce biases in the downstream analysis. More specifically, reads underwent hard trimming from the 5′ end with Trim Galore to match

the shortest read length in the dataset (50 bp), and only the first mate of paired-end reads was used to ensure compatibility between single- and paired-end samples. ATAC-Seq and CUT&RUN were subsequently aligned to the human genome hg38 with bowtie2 (v2.4.1)[144] using default parameters. The hard trimming of CUT&RUN reads reduced the average alignment rate by only 4% when compared to a default adapter trimming with Trim Galore, thus indicating that the additional filtering did not result in large loss of data. Next, reads which were unmapped, duplicate, with MAPQ < 30, or mapping to chromosomes other than chr1-22, X or Y were removed using sambamba (v0.5.6)[132]. For ATAC-Seq samples, peaks were called for each developmental stage using Genrich (v0.6, https://github.com/jsh58/Genrich) with the following parameters: -q 0.01 -j -y -v. For CUT&RUN samples, peaks were called for each developmental stage using MACS2 (v2.1.0)[145] with the following parameters: --broad –broad-cutoff 0.05 -q 0.05 -g hs. Consensus and differential peaks across developmental stages were calculated with DiffBind (v3.0.8)[146], using the summits=500 parameter for ATAC-Seq and summits=1000 parameter for CUT&RUN datasets. Enrichment analysis of peaks overlapping TSSs was performed using fgsea (v1.16.0)[147]. Normalization factors for each sample were estimated by applying the calcNormFactors function from EdgeR (v3.32)[148] with method "TMM" to a matrix containing read counts across all consensus peaks. Scaling factors were next identified by multiplying the normalization factors by the total number of reads mapped across all peaks for each sample divided by a factor of $10^6$, and subsequently taking the reciprocal of the resulting value. BigWig files were generated using bamCoverage from deepTools (v3.5.0)[149], by providing the scaling factors calculated for each sample using the --scaleFactor parameter. Lastly, coverage for each developmental stage was calculated by averaging the signal across scaled replicates for each developmental stage using the mean function from wiggletools (v1.2)[150]. Genomic regions ±500 bp of TSSs from the transcriptome were defined in an R 4.0.3 environment. Random genomic 1000 bp regions were generated using bedtools (v2.29.2)[151] shuffle, by providing a BED file of the genomic locations of transcript locations from Ensembl v102 and chromosome gaps to the -excl parameter. Normalized pileup at TSS regions and shuffled background locations was calculated using deepTools using the computeMatrix reference-point function with the following parameters: --referencePoint center --beforeRegion-StartLength 500 --afterRegionStartLength 500. Density profiles were plotted using the plotHeatmap function.

### Analysis of publicly available RRBS data

Publicly available human embryo RRBS data was downloaded from the SRA database using accession number SRP028804. Raw RRBS reads underwent adapter trimming and quality control with Trim Galore (v0.6.6) using the following parameters: --rrbs --paired. Next, Bismark (v0.22.3)[152] was used to align trimmed reads to the human genome hg38 and generate a genome-wide report of cytosine methylation in the CpG context with default parameters. DNA methylation at regions ±500 bp of TSSs and at random genomic 1000 bp regions (as defined above) was calculated using methylKit (v1.16.0)[153]. For each region, the percentage of DNA methylation was defined by dividing the number of identified Cs (methylated reads) by the total number of identified Cs and Ts (unmethylated reads) within the region. Lastly, average methylation levels for each region were identified by calculating the arithmetic average across replicates in each developmental stage.

### Isoform coordinate lifting and quantification in non-human genomes

Isoform genomic coordinates were converted from the human hg38 genome to other vertebrate genomes using the liftOver (v9-Jul-2019)[67] utility from UCSC with default parameters. Lifting was performed by using chain files for the following genomes: chimpanzee (PanTro6), rhesus macaque (RheMac10), marmoset (CalJac4), pig (SusScr11),

mouse (Mm10), chicken (GalGal6), and zebrafish (DanRer11). Publicly available RNA-Seq data generated from rhesus macaque and marmoset preimplantation embryos were downloaded from the SRA and ENA databases from the following studies: SRP089891 (Wang et al., macaque embryo RNA-Seq) and PRJEB29285 (Boroviak et al., marmoset embryo RNA-Seq). To assess expression of novel isoforms in these species, the genePred files containing fully mapped isoforms generated by liftOver were converted to GTF using the genePredToGtf utility from UCSC, and subsequently used to generate STAR and RSEM indices with the corresponding genome assembly nucleotide sequences. Raw RNA-Seq reads underwent adapter trimming, quality control, and were subsequently aligned to the respective reference genomes and used to quantify isoform expression using the newly generated indices as described above for the human samples.

### Differential gene and isoform expression analysis

Gene- and isoform-level raw counts generated by RSEM were imported into an R 4.0.3 environment using tximeta (v1.8.2)[154]. Principal Component Analysis was performed using the prcomp R function on a matrix containing expression levels of the top 2500 most variable genes following normalization with the varianceStabilizing-Transformation function from DESeq2 (v1.3.0). Per-stage distributions of isoform novelty were calculated by first taking genes with >10 counts in each sample, and subsequently calculating the arithmetic average of the percentage of short RNA-Seq reads mapping to novel isoforms for each gene and stage. Differential gene expression analysis across developmental stages was performed using DESeq2 (v1.3.0)[155] using default parameters. P-values were corrected using the Benjamini-Hochberg method. Differentially expressed genes were defined as having an adjusted p-value ≤ 0.05 and log2FoldChange > 1 (upregulated) or log2FoldChange < -1 (downregulated). Differential isoform expression analysis was performed using kallisto (v0.46.1)[156] (run with –bootstrap-samples 100) and sleuth (v0.30.0)[70]; significant isoforms were filtered by requiring an adjusted p-value ≤ 0.05 and beta value > 1.5 (upregulated) or beta value < -1.5 (downregulated). Enrichment analysis of novel isoform classes was performed on isoform-level differential expression signatures for each developmental stage transition using fgsea (v1.16.0)[157]. Isoform plots in Supplementary Fig. 4C–E were generated using IsoformSwitchAnalyzeR[158].

### Alternative splicing analysis

Alternative splicing across developmental stages was profiled using SUPPA2 (v2.3)[87]. First, the novel transcriptome GTF file and transcript TPM values from RSEM were used to measure the percent spliced in (PSI) values for isoforms and seven types of alternative splicing events for each sample: skipped exon (SE), alternative 5' splice site (A5), alternative 3' splice site (A3), retained intron (RI), mutually exclusive exons (MX), alternative first exon (AF), alternative last exon (AL). Significant isoform switching and alternative splicing events across consecutive developmental stages were next identified by using the SUPPA2 diffSplice function. Significant AS events were defined as having a p-value < 0.05 and differential PSI > 0.1 (more included) or < -0.1 (less included). Gene Ontology enrichment analysis of genes associated with at least one significant AS event per comparison was performed using the gprofiler2 (v0.2.0) R package. Isoforms were clustered based on their relative inclusion across developmental stages by providing the average PSI value for each stage to the Mfuzz (v2.50.0)[159] R package. Only isoforms with a statistically significant switching event in at least one comparison were used for the analysis. Statistical significance between distributions of coding probability, evolutionary conservation scores, intron number and length across clusters were calculated using the Wilcoxon rank-sum test in an R 4.0.3 environment and adjusted using the Benjamini-Hochberg method. Correlation between mRNA expression of SFs (defined as genes annotated in the Gene Ontology Biological Process term "RNA

splicing", GO:0008380) and isoform PSI across short-read RNA-Seq replicates was calculated using the cor.test function in R with the Spearman method and pairwise complete observations parameter. SF eCLIP peak files (BED format) and coverage files (BigWig format) were downloaded from the ENCODE database[46]. Isoform-SF pairs in the network (Supplementary Fig. 5D) were filtered by requiring an absolute correlation value greater than 0.75, and the presence of at least one eCLIP SF binding peak overlapping the isoform primary sequence on the same strand.

## Weighted gene co-expression network analysis

Gene co-expression networks were generated from gene-level expression data across all short-read RNA-Seq samples across developmental stages in an R 4.0.3 environment using WGCNA (v1.69)[95]. Genes were first filtered by requiring >10 counts across all samples. Raw expression counts were subsequently normalized using size factors from DESeq2 (v1.3.0), and lastly transformed by performing $\log_{10}(x+1)$. A signed adjacency matrix was calculated from gene expression data using a power of $\beta = 13$ and converted to a signed topological overlap matrix, which was used to perform gene clustering with the hclust function using the "average" clustering method. Modules were defined by cutting the clustering tree using the Dynamic Hybrid Tree Cut method with a minimum cluster size of 30 genes. Modules whose eigengenes had a Pearson correlation ≥0.95 were merged. Gene Ontology enrichment analysis was performed for each module using the gprofiler2 (v0.2.0)[160] R package and the Gene Ontology: Biological Process library. miRNA-gene binding was predicted using miRanda (v3.3a)[100] with default parameters, using miRNA sequences from miRbase (v22.1)[161]. miRNA-gene pairs were filtered by requiring an alignment in the 3' UTR region of at least 10 bp and at least 85% sequence identity (allowing wobble base pairing). Over-representation analysis was performed using clusterProfiler (v3.18.0)[162], with a maximum gene set size of 5000, and filtered using $p$-value < 0.05, adjusted using the Benjamini-Hochberg method. The genome-wide co-expression network was plotted with ggnetwork (v0.5.10) using the Fruchterman-Reingold layout after removing edges with ≤0.1 connectivity in the signed TOM matrix generated from WGCNA and resulting unconnected nodes. Module preservation analysis was performed using the modulePreservation WGCNA function, computing a $Z_{summary}$ and $p$-value (Bonferroni correction) for each module representing the preservation of the module's network topology across each independent dataset. For the two primate datasets, the analysis was performed using the human reference transcriptome lifted to the respective species genome using liftOver and quantified using the short-read RNA-Seq embryonic samples.

## Novel gene cluster analysis

Novel gene clusters were defined by applying Mfuzz (v2.50.0) to a matrix containing gene expression levels normalized using the varianceStabilizingTransformation function from DESeq2 (v1.3.0). Motif enrichment analysis for each novel gene cluster was performed with HOMER (v4.10) on promoter regions defined between 3000 bp upstream and 500 bp downstream of each gene's major isoform TSS, using the -size given parameter and other novel gene cluster promoters as background. Predicted motifs were further filtered by requiring an enrichment $q$-value < 0.05, and the corresponding TF to bind at least 10 novel gene promoters and have valid predicted TF activity scores (see below for details). The top 5 results for each novel gene cluster are displayed in Fig. 4D. Gene Ontology enrichment analysis of novel gene clusters was performed using gprofiler2 (v0.2.0) by submitting the top 1000 most connected known genes for each cluster, as ranked by average connectivity from the WGCNA TOM matrix weighted by each gene's Mfuzz cluster membership.

## Construction of the TF-novel gene regulatory network

TF-novel gene regulatory interactions were predicted by integrating TOBIAS (v0.11.1)[110] and VIPER (v1.24.0)[111]. First, BAM files generated by alignment of ATAC-Seq replicates for each developmental stage (from Liu et al.) were merged, and Tn5 insertion bias was corrected using ATACorrect by providing previously defined peaks (see above) and the hg38 blacklist file from ENCODE[163] (https://github.com/Boyle-Lab/Blacklist/). Next, FootprintScores was used to estimate TF footprint scores, which were used by BINDetect in combination with non-redundant TF motifs from JASPAR CORE (9th release)[164] to predict bound TFs for each developmental stage. Putative TF-target regulatory interactions were determined by identifying TFs predicted to be bound at novel gene promoters (3000 bp upstream and 500 bp downstream of each isoform TSS). Next, TF activity was predicted for each short-read RNA-Seq sample using VIPER, using human regulons from DoRothEA (v1.2.2)[165] and gene expression values normalized as for the network analysis. The TF-target network was further refined by requiring the predicted TF activity and the novel gene's normalized expression to have absolute correlation values ≥ 0.3 across short-read RNA-Seq replicates, as determined by Spearman's index, as an adjusted $p$-value < 0.05 (Benjamini-Hochberg correction).

## PCR validation of novel genes

PCR primers were generated using NCBI Primer-BLAST[166] and checked for off-target effects against the genome using UCSC in silico PCR (https://genome.ucsc.edu/cgi-bin/hgPcr). To capture a diversity of isoform structures, two pairs of primers were designed for each gene: one pair that encompasses the most common outer pair of exons within the gene, and a second pair that captures at least one internal exon (see Supplementary Fig. 8A, full primer sequences are available in Supplementary Data 9). RNA was extracted and reverse transcribed to cDNA as described above. Samples were collected from two early preimplantation stages: day 1 (1C) embryos and day 3 (8C) embryos, and two biological replicates were generated for each developmental stage. As a result, four separate samples were assessed: 1C-1 and 1C-2 (both independently generated by pooling three sets of separate 1C embryos), 8C-1 and 8C-2 (pooling four E3 embryos each). To allow for detection of genes with varying levels of expression, cDNA was amplified by PCR using 30–36 cycles, and gel lanes were loaded with 10–200 ng cDNA.

## Statistics and reproducibility

IVF embryos used in this study were donated for research by patients following informed consent and after completing their family fertility plan. In order to ensure reproducibility of our experimental findings, we extracted RNA from high-quality embryos as assessed with the scoring system routinely used in IVF clinical cycles. Nevertheless, for some stages (i.e. morula) we did not have a large enough selection of embryos from our banked resource, and used a limited number of morula stage embryos as noted within the methods and results accordingly. Regardless, our embryo collection is a sufficient presentation of IVF preimplantation embryos representing the natural variability in the population across the zygotic to blastocyst stages of development. No statistical method was used to predetermine sample size but we included 13 to 16 embryos at all stages of development with the exception of $N = 3$ at the morula stage as illustrated in Fig. 1A. Experiments were not randomized and investigators were not blinded to allocation during experiments and outcome assessment, as this was not considered relevant to the study.

## Reporting summary

Further information on research design is available in the Nature Portfolio Reporting Summary linked to this article.

## Data availability

The long- and short- RNA-Seq sequencing data have been deposited at GEO under the accession GSE190548. Data can be also interactively explored and downloaded from the following custom built web resource: https://denis-torre.github.io/embryo-transcriptome/. Predicted biological properties for isoforms in the transcriptome are provided as Supplementary Data. Source data are provided with this paper.

## Code availability

All analyses were performed using publicly available tools in bash 4.2.46, Python 3.8.2 and R 4.0.3 environments. The code has been deposited on GitHub at the following link https://github.com/denis-torre/embryo-transcriptome, and can be cited via Zenodo at https://doi.org/10.5281/zenodo.8368062.

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

## Acknowledgements

This work included support from the Sagol Network awarded to DBY. The authors thank the dedicated team of embryologists and medical professionals at the Institution of Reproduction and IVF, Lis Maternity Hospital, Tel Aviv Sourasky Medical. We would like to deeply appreciate and thank the consented IVF patients of the Tel-Aviv Medical Center who appreciate our clinical and scientific research and donated their spare IVF embryos after completing family planning. We also want to thank Jill Gregory, CMI FAMI, Associate Director of Instructional Technology at the Icahn School of Medicine at Mount Sinai for creating the illustration used in Fig. 1A. Icons in Figs. 2A, 2C, 3C, and 8A–D were created with BioRender.com.

## Author contributions

R.S., D.B.Y. conceived, designed and funded the study. D.T., N.J.F., D.B.Y and R.S. wrote the initial draft. D.T. and N.J.F. developed the computational pipeline and conducted the analysis. Y.K. performed IVF. I.G.C. performed RNA extraction and library preparation for sequencing. N.J.F., B.S.M., G.D., K.A., K.V., K.M. and M.L.S. coordinated and performed the RNA sequencing. H.S., Y-C.W., S.H.S. provided bioinformatics support. R.F. and M.F. performed PCR validation and designed primers. E.E., F.A., H.A., Y.M., I.M., M.L.S., E.G. and E.S. provided crucial intellectual contribution. All listed authors provided input into the manuscript editing and revisions and approved its final form.

## Competing interests

R.P.S. is an equity holder and paid consultant to GeneDx. However, the research was solely conducted by MSSM and TASMC facilities. The remaining authors declare no competing interests.

## Additional information

[1]Department of Genetics and Genomic Sciences, Icahn School of Medicine at Mount Sinai, New York, NY 10029, USA. [2]Pacific Biosciences, Inc., Menlo Park, CA 94025, USA. [3]Fertility and IVF Institute, Tel-Aviv Sourasky Medical Center, Affiliated to Tel Aviv University, Tel Aviv 64239, Israel. [4]Center for Advanced Genomics Technology, Icahn School of Medicine at Mount Sinai, New York, NY 10029, USA. [5]Icahn Genomics Institute, Icahn School of Medicine at Mount Sinai, New York, NY 10029, USA. [6]Department of Cell and Developmental Biology, Sackler Faculty of Medicine, Sagol School of Neuroscience, Tel-Aviv University, Tel-Aviv 69978, Israel. [7]CORAL – Center Of Regeneration and Longevity, Tel-Aviv Sourasky Medical Center, Tel Aviv 64239, Israel. [8]Immunai Inc., New York, NY 10016, USA. [9]Department of Biochemistry and Molecular Genetics, University of Louisville, Louisville, KY 40202, USA. [10]Department of Molecular Cell Biology, Weizmann Institute of Science, 7610001 Rehovot, Israel. [11]Department of Biological Chemistry, Center for Epigenetics and Metabolism, University of California, Irvine, CA 92697, USA. [12]Center for OncoGenomics and Innovative Therapeutics (COGIT); Department of Oncological Sciences, Icahn School of Medicine at Mount Sinai, New York, NY 10029, USA. [13]Black Family Stem Cell Institute, Icahn School of Medicine at Mount Sinai, New York, NY 10029, USA. [14]These authors contributed equally: Denis Torre, Nancy J. Francoeur, Yael Kalma. [15]These authors jointly supervised this work: Dalit Ben-Yosef, Robert Sebra. ✉e-mail: dalitb@tlvmc.gov.il; robert.sebra@mssm.edu

