## [Peer Review File · Nature Communications]

REVIEWER COMMENTS

Reviewer #1 (Remarks to the Author):

This is an outstanding manuscript that provides the first and much needed isoform resolved RNA-seq analysis of different stages of human embryo development. Not only it provides a critical resource, but it reaches several novel findings.

I have reviewed this manuscript previously at one of the top Cell press journals, and unfortunately this paper was rejected after revision, although the authors have addressed all my comments and in my opinion, all other reviewers comments.

Now I am asked to review this work in Nature Communications, and the current version is basically the revised version which has been drastically improved, and already addressed my previous comments.

The conclusions are fully supported by the data, and the figures are elegant and coherent. The data is of very high quality and is extensive.

I have no requests for other improvements and support publication of the current submitted version.

Reviewer #2 (Remarks to the Author):

Comments to the Author

Torre et al. profile the transcriptome of the human embryo in the isoform level using bulk RNA-seq, long-read sequencing, and single-cell RNA sequencing. The authors declared this to be the first article using these sequencing data to profile the human embryo at the isoform level. Novel isoforms are supported with public multi-omics data. Abundant novel transcriptome resources are also presented online which is helpful to others.

Major suggestions:

1. The authors investigated human preimplantation development using a lot of computational approaches to characterize the isoform-resolved transcriptome and found distinct patterns of its properties, such as protein-coding potential, transposable element content, evolutionary conservation,

transcriptomic and epigenomic landscape, differential gene and isoform expression, alternative splicing, and gene co-expression. In the discussion section, it would be better to integrate them to give a more comprehensive summary and deeper insight of all their findings, which will increase the value of this work for future research.

2. The authors presented three novel genes identified as examples. Although multiple-omics data were used to support the validation of these novel genes, it is recommended to use PCR or other experimental methods to validate novel isoforms.

3. To my knowledge, DESeq2 is designed to analyze expression differences in gene level. But the authors used DESeq2 to perform the differential analysis in the isoform level with default parameters. It is suggested to use other methods designated for transcript-level/isoform-level differential analysis.

4. The authors performed an integration of bulk RNA-seq and single-cell RNA-seq based on the novel transcriptome reference. However, this integration seems to contribute little to the confidence of novel transcripts. Can authors explain in detail how the Figure. 4D shows the confidence of novel transcriptome. Moreover, the difference between known and novel transcriptome reference used in the integration may be interesting. Is it possible that a novel transcriptome in high-resolution can contribute to the cell clustering?

Minor suggestions:

1. The conclusion in Figure. 1D that the coding potential is positively associated with the isoform length is not convincing. Based on the scatter plot provided, the author suggests a positive correlation between protein-coding potential and length, but this conclusion is deemed unreliable. The fit of the scatter plot can lead to different results depending on the function chosen, such as a segmented function. Furthermore, this conclusion lacks biological significance. It is recommended that the author adjust the interpretation of these results.

2. In Figure 2C, we can see the distinct patterns of isoform expression throughout all embryonic stages across all datasets, but the difference between isoform classes cannot be seen obviously from the picture.

3. The authors described the TF-novel gene interaction network is “high-confidence” in line 564-565. How is it concluded?

4. In Figure. 3A, the signals of H3K4me3 and H327ac around novel antisense/intergenic gene TSS are weak. CAGE-seq may be helpful to validate the existence of these novel TSSs.

5. What is y-axis of the curves in Figure 4B, the distribution density? If the x-axis represents the percentage of short RNA-Seq reads, why the percentage may be less than 0 or more than 100%.

6. The cluster tree of the hierarchical clustering in some heatmaps such as in Figure 7C, D doesn't make any sense except sorting, and it is not necessary to display the meaningless cluster tree.

7. The figure legend of Figure 8B, C (Line 1157-1160) need to be improved.

Reviewer #3 (Remarks to the Author):

Torre et al present a valuable resource of isoform-resolved transcriptome of the human preimplantation embryo. The data clearly highlights the underestimated complexity of the human transcriptome at the earliest stages of development especially during the period from zygote to zygotic genome activation. I therefore recommend publication following minor revision.

The novel isoforms detected during early human embryo may be maternally loaded transcripts. It would be interesting to specifically examine this in previously published datasets of oocytes. doi: 10.1038/ncomms9207 contain a large number of oocytes and may be a useful dataset to include for such analysis.

In Figure 2 D I'm not sure the Mazid et al. integration is performed in the correct manner as the primed cells looks like they are mixing with the TE cells of Petropoulos et al. Also, the 8CLC cluster with E4 cells of Petropoulos et al and not with the E3. How significant is the increased gene set score for the Mazid et al, considering it is -0.1 in the 8CLC which is lower than in the E3 in Petropoulos et al? Could the low score be explained by the possibility that many of these transcripts are maternally loaded?

Could also the relatively "silent" epigenetic signatures for "Novel antisense gene TSS" and "Novel intergenic gene TSS" in Figure 3A, B also be explained by maternal loading without active transcription at later stages?

Reviewer #1 (Remarks to the Author):

This is an outstanding manuscript that provides the first and much needed isoform resolved RNA-seq analysis of different stages of human embryo development. Not only it provides a critical resource, but it reaches several novel findings.

I have reviewed this manuscript previously at one of the top Cell press journals, and unfortunately this paper was rejected after revision, although the authors have addressed all my comments and in my opinion, all other reviewers comments.

Now I am asked to review this work in Nature Communications, and the current version is basically the revised version which has been drastically improved, and already addressed my previous comments.

The conclusions are fully supported by the data, and the figures are elegant and coherent. The data is of very high quality and is extensive.

I have no requests for other improvements and support publication of the current submitted version.

We thank the Reviewer for appreciating the novelty and impact of our work, and agree that we placed a tremendous amount of additional information into this version of the manuscript. Not only does the manuscript supply the most comprehensive human preimplantation embryo transcriptome reference, but also serves to showcase both known and novel gene- and isoform-level expression across the catalog of published multi-omics and recent model system datasets. This further motivates additional functional studies in human development, alongside providing a publicly accessible portal and database for leveraging the raw data, reference set, and in silico predictions.

Reviewer #2 (Remarks to the Author):

Comments to the Author

Torre et al. profile the transcriptome of the human embryo in the isoform level using bulk RNA-seq, long-read sequencing, and single-cell RNA sequencing. The authors declared this to be the first article using these sequencing data to profile the human embryo at the isoform level. Novel isoforms are supported with public multi-omics data. Abundant novel transcriptome resources are also presented online which is helpful to others.

Major suggestions:

1. The authors investigated human preimplantation development using a lot of computational approaches to characterize the isoform-resolved transcriptome and found distinct patterns of its properties, such as protein-coding potential, transposable element content, evolutionary conservation, transcriptomic and epigenomic landscape, differential gene and isoform expression, alternative splicing, and gene co-expression. In the discussion section, it would be better to integrate them to give a more comprehensive summary and deeper insight of all their findings, which will increase the value of this work for future research.

We thank the Reviewer for the suggestion and agree on the value of integrating this additional content. We have now extensively reworked the Discussion section to better illustrate the key novel findings of our manuscript, and explain more in depth how the multiple computational approaches we applied come together to help us reach those conclusions. We believe the Discussion is significantly strengthened in this new version, and will improve the utility of this work for users seeking to leverage our data resource in the future.

Below is a brief summary of the main aspects we have expanded on:

1. **Characterizing novel genes:** we added context to further explain how our predictions and analysis results are integrated to characterize the novel antisense and intergenic genes we report herein. These genes are largely non-coding, rich in hominid-specific TEs, poorly evolutionarily conserved, and largely expressed in earlier preimplantation stages – typically either maternally inherited or transiently expressed during EGA.
2. **Novel alternative splicing events of known genes:** in addition to novel genes, our predictions also inform the many novel isoforms detected for known genes. We provide extensive predictions of properties for each isoform, which will be useful on a case-by-case basis. Nonetheless, our analyses also revealed some global patterns of splicing. Notably, we find that EGA is associated with transient inclusion of unannotated, non-coding, and poorly conserved isoforms.
3. **Custom built web server:** we also highlight the custom-built web server to interactively access, explore, and download the full set of isoforms and associated predictions described in this study, <https://denis-torre.github.io/embryo-transcriptome/>. This is intended to be a highly accessible and utilized portal for using our resource for integration into community studies.

2. The authors presented three novel genes identified as examples. Although multiple-omics data were used to support the validation of these novel genes, it is recommended to use PCR or other experimental methods to validate novel isoforms.

To orthogonally support the existence of the novel genes illustrated in **Figure 8**, we have now performed PCR validation using a new set of embryos, and included the results in the new **Supplementary Figure 8** (see below). To further increase confidence in the novel genes, we also performed additional in-silico validation using independently published short-read RNA-Seq data, now displayed in **Supplementary Figure 9** (see below, after PCR results).

First, we generated PCR primers using the NCBI Primer-BLAST design tool (PMID 22708584, <https://www.ncbi.nlm.nih.gov/tools/primer-blast/>), and checked for off-target effects against the genome using UCSC in silico PCR (<https://genome.ucsc.edu/cgi-bin/hgPcr>). To capture a diversity of isoform structures across novel genes, we designed two pairs of primers for each gene: one pair that encompasses the most common outer pair of exons within the gene, and a second pair that captures at least one internal exon (in some cases a minor exon). The specific locations of these primers with respect to gene models are shown below, together with the expected length of PCR products for each primer set (**Suppl. Figure 8A**):

Supplementary Figure 8A. Isoform models and repetitive element annotations (RepeatMasker) for NOVELG000067783 (G1), NOVELG000084291 (G2), NOVELG000070644 (G3), and NOVELG000059671 (G4). Also highlighted are the genomic locations of the forward and reverse primer pairs with corresponding expected length of the PCR product. Note: the reverse primer for G2-p2 maps to both the last exon (light blue, left-most in the panel) and an internal exon, due to the presence of repetitive elements (LTR7); in-silico PCR does not predict other off-target effects.

Next, we extracted RNA from a new set of early human preimplantation embryos at two separate stages to best demonstrate expression of novel genes: day 1 (1C) embryos, and day 3 (8C) embryos, with two biological replicates for each developmental stage. We subsequently generated cDNA, performed PCR using the primer pairs described above, and visualized the bands via gel electrophoresis. Notably, we were able to confirm expression of all four novel genes (and multiple separate isoforms) across independent replicate samples and runs (see **Suppl. Figure 8B-I** below):

Supplementary Figure 8B-I. Gel electrophoresis displaying PCR results from human preimplantation embryo cDNA samples, using primer pairs illustrated in **Suppl. Figure 8A**. Four separate samples were assessed: 1C-1 and 1C-2 (both independently generated by pooling three sets of separate 1C embryos), 8C-1 and 8C-2 (pooling four E3 embryos each).

A summary of the detection status for each gene and primer pair is displayed below, alongside relevant notes (**Reviewer Table 1**):

Gene	Primer pair	Validated by PCR	Validated by independent RNA-Seq data	Notes
G1	p1	Yes (B, C, D, I)	Yes	Strong bands in 1C and 8C, consistent with high expression in RNA-Seq
	p2	No	Yes	Lowly expressed, likely needs more cDNA/PCR cycles for detection
G2	p1	Yes (E)	Yes	Detected in 1C but not 8C, consistent with downregulation in RNA-Seq
	p2	Yes (B, H)	Yes	
G3	p1	No	Yes	Lowly expressed, likely needs more cDNA/PCR cycles for detection
	p2	Yes (H, I)	Yes	Detected in 1C, some off-target effect in one gel possibly due to repetitive elements
G4	p1	Yes (D, E, F, G)	Yes	Detected in 1C and 8C, weak bands consistent with low expression in RNA-Seq
	p2	No	Yes	Lowly expressed, likely needs more cDNA/PCR cycles for detection

Reviewer Table 1. Summary of the validation status by PCR and sequencing for each gene and primer pair profiled. Letters in the “Validated by PCR” correspond to matching panels displaying detected bands in **Suppl. Figure 8**.

Out of all genes and primer pairs tested, NOVELG000067783 (G1) was observed to be the most highly expressed by PCR, consistently with the RNA-Seq data. We were also able to show bands within expected length ranges for both primer pairs for G2, as well as one primer pair each for G3 and G4, all of which were more lowly expressed, consistently with RNA-Seq data.

While we successfully physically validated the majority of primer pairs we attempted (and all genes using independent NGS data, shown below after the PCR results), not all primer pairs yielded visible bands on the gels, likely due to one or more of the following factors:

- Due to the limited number of spare human embryos donated for research purposes at these early stages, calibration of the PCR reaction conditions and primers was initially performed on RNA extracted from human embryonic stem cells, representing the closest developmental proximity to the blastocyst stage. However, it should be noted that most of the profiled gene expression levels are low to non-existent in blastocyst based on RNA-Seq data, which limited the efficiency of these calibration experiments. Rather, the housekeeping gene GAPDH was used as a positive control, and pure water as a negative control.
- Due to the above, the bands displayed in **Suppl. Figure 8** include different combinations of parameters that were gradually optimized to allow for detection of lowly expressed genes. For example, in one of our first iterations (**Suppl. Figure 8B**, using 32 PCR cycles and 10ng cDNA per well), we were able to detect some bands (e.g. G1-p1, G2-p2) but not others (e.g. G2-p1). We subsequently repeated the experiment from the same sample with additional PCR cycles and cDNA, which allowed us to identify bands that we had previously not detected (**Suppl. Figure 8E**, 34 cycles, 30ng cDNA).
- Aside from G1, all novel genes profiled are rich in repetitive elements and low-complexity regions. While PacBio SMRT-Seq can fully span and accurately sequence these elements, PCR primer design at such regions is challenging, and we had to customize parameters in the Primer-BLAST design tool to obtain usable primers (i.e. low-complexity and repeat filters given the complexity of the novel events). While we overall detected expected bands for the tested primers, it is possible that some of the extra bands for G3-p2 in **Suppl. Figure 8H** (appearing above the expected 217 bp band) might be due to such repetitive elements, as G3 is especially rich in LTR7 repeats, which the primers overlap.

In summary, the experiments above show that all four novel genes and multiple isoforms are supported by PCR. Additionally, to further increase confidence in these genes, we have integrated short-read RNA-Seq datasets from independent studies profiling human preimplantation embryos across multiple stages, all of which show clear support for the novel genes and associated isoforms, with consistent patterns of expression across stages (new **Supplementary Figure 9**, see below):

Supplementary Figure 9. Short-read RNA-Seq data from independently published human preimplantation embryo studies shows support of all four novel genes highlighted.

3. To my knowledge, DESeq2 is designed to analyze expression differences in gene level. But the authors used DESeq2 to perform the differential analysis in the isoform level with default parameters. It is suggested to use other methods designated for transcript-level/isoform-level differential analysis.

We thank the Reviewer for this valuable suggestion. While the DESeq2 developers have shown that differential gene expression methods can be applied to transcript-level differential analysis with reasonably good performance (PMID 30356428), the Reviewer is indeed correct in stating that there are other methods that have been specifically designed for this purpose, which achieve superior performance and better account for quantification uncertainty in transcript-level expression estimates.

We have thus repeated the transcript-level differential expression analysis by using kallisto (PMID 27043002) and sleuth (PMID 28581496), which have specifically been designed and tested for this purpose. More specifically, we first ran kallisto by performing 100 bootstrap samples (i.e. using the *--bootstrap-samples=100* parameter), which allows to quantify inferential variance for transcript-level expression estimates. Next, we ran sleuth, which leverages these bootstrap estimates to perform differential transcript analysis across pairwise comparisons, accounting for multiple experimental conditions. Lastly, we counted the significantly upregulated ($q\text{-value} < 0.05$, $\beta\text{ value} > 1$) and downregulated ($q\text{-value} < 0.05$, $\beta\text{ value} < -1$) transcripts at each pairwise stage transition, similarly to the previous analysis. Results are included in the revised **Figure 4C**:

Figure 4C. Number of significantly differentially expressed isoforms at each developmental stage transition.

The results confirm that the 4C vs 8C stage transition continues to show the largest changes in isoform expression, consistently with the results found using DESeq2. We note that the overall number of significantly differential isoforms is reduced, potentially suggesting a better control of the false discovery rate (FDR), as suggested by the authors (compared to other methods including DESeq2, PMID 28581496). Nonetheless, the interpretation of the data remains unchanged. Of note, we additionally observed an increase in the number of upregulated isoforms in the morula vs blastocyst transition, consistent with early lineage differentiation. While our intent with this Figure was to catalog the relative temporal isoform expression, we will certainly make the raw

data available on the manuscript data portal as we appreciate there are several different parameters and methods that can be used to analyze these data, given there is no singular method used in the community today. Our methods were updated to highlight these analyses.

We also repeated the isoform-level enrichment analysis in **Suppl. Figure 4A** using differential expression estimates from sleuth, which shows a significant negative enrichment of novel antisense and intergenic isoforms at developmental stage transitions most strongly at the 4C vs 8C stages and beyond, consistent with the results previously generated using DESeq2:

Suppl. Figure 4A. Enrichment analysis of novel antisense and intergenic isoforms along isoform-level differential expression signatures across developmental stages. Isoforms are ranked by most strongly upregulated (lowest rank, on left of each subplot) to most strongly downregulated (highest rank, on right). Vertical bars represent the position of genes of each class within the ranking. Also reported are p-values and normalized enrichment scores (NES).

4. The authors performed an integration of bulk RNA-seq and single-cell RNA-seq based on the novel transcriptome reference. However, this integration seems to contribute little to the confidence of novel transcripts. Can authors explain in detail how the Figure. 4D shows the confidence of novel transcriptome. Moreover, the difference between known and novel transcriptome reference used in the integration may be interesting. Is it possible that a novel transcriptome in high-resolution can contribute to the cell clustering?

We thank the Reviewer for the comments and suggestions for additional analyses to explore the contribution of novel isoforms to the clustering of early embryonic samples.

With regards to the first question, we assume the Reviewer is referring to **Figure 2D**, which displays UMAP plots of integrated single-cell RNA-Seq datasets (**Figure 4D**, which displays the expression patterns and isoform novelty across known developmental genes, aims to underscore that there is much isoform novelty across known regulators of embryo development). The purpose of the panel was intended to give readers an overview of the integrated datasets and illustrate the clustering of cells quantified against our isoform-resolved reference, underscoring that model systems such as 8CLC validate the reference when quantified against it; however, the panel itself does not provide direct information on the support of the novel isoforms.

Following these comments, as well as feedback from Reviewer #3, we have now restructured the analysis of scRNA-Seq data to better underscore how it contributes to the confidence of the novel transcriptome, and to demonstrate how novel isoforms can contribute to cell clustering. First, we removed previous **Figure 2D**, which primarily served to give an overview of the integrated datasets, but also relied on known isoform annotations; as well as **Figure 2E**, which displayed the progression of novel gene expression, but had been quantified using a gene set approach whose values are not directly meant for comparison across datasets. We have now added improved panels to **Figure 2** and **Suppl. Figure 2**, which we believe are better suited to explaining how these datasets support our findings.

First, we have added additional bar plots to **Figure 2C** (see below), which display the detection level of the isoform classes across stages in Petropoulos et al. (previously in the supplement), as well as two additional short-read RNA-Seq datasets containing large amounts of oocyte and 1C samples (Asami et al, PMID 34936886, and Tohonen et al., PMID 26360614, following requests from Reviewer #3). These results show that all classes of novel isoforms are widely expressed across stages in Petropoulos et al. (primarily E3), with lower detection levels in later stages (e.g. E7), consistently with observations from our and other datasets.

We have also added bar plots displaying detection levels of isoform classes across clusters in the single-cell RNA-Seq data from Mazid et al. and Kagawa et al. to **Figure 2D** (see below, previously in the supplement), which more clearly show the extent of expression of these isoforms in in-vitro models to investigate human preimplantation development. These results indicate that the novel isoforms are also broadly detected in these models, though at overall lower levels than those of human embryos, possibly due to the fact some of these are maternally inherited and thus not reactivated in such models.

Figure 2C. Percentage of isoforms in each developmental stage, grouped by isoform class and average expression level across short-read RNA-Seq datasets profiling human preimplantation embryos and oocytes (TPM - Transcript Per Million) **Figure 2D** displays data from single-cell short-read RNA-Seq datasets (SmartSeq2) profiling in-vitro models of human preimplantation development (8CLCs and blastoids).

We additionally sought to address the question of whether the novel transcriptome can contribute to cell clustering in these single-cell RNA-Seq datasets, as asked above. First, we calculated isoform-level expression for individual cells in each dataset (using STAR alignment, followed by RSEM quantification). This was made possible because these datasets were generated using the SmartSeq2 technology, which results in read coverage across the entire gene body (unlike other technologies such as 10X, which typically only generate read pileups at the 3' or 5' ends of the gene). We then generated UMAPs on normalized expression data calculated exclusively from expression of novel isoforms. Notably, we found that novel isoforms alone are able to effectively separate cells along their developmental trajectory, or respective cell clusters, across all profiled datasets (now included in **Suppl. Figure 3C**):

Suppl. Figure 3C. UMAP plots generated by quantifying gene expression of the orthogonal, previously published datasets to the novel isoforms in the isoform-resolved transcriptome presented in this study.

We thank the Reviewer again for this suggestion, and we believe that the updated Figure better communicates that the novel isoforms reported in this study are widely supported and can contribute to separating developmental stages and conditions across these multiple independently published studies.

Minor suggestions:

1. The conclusion in Figure. 1D that the coding potential is positively associated with the isoform length is not convincible. Based on the scatter plot provided, the author suggests a positive correlation between protein-coding potential and length, but this conclusion is deemed unreliable. The fit of the scatter plot can lead to different results depending on the function chosen, such as a segmented function. Furthermore, this conclusion lacks biological significance. It is recommended that the author adjust the interpretation of these results.

We thank the Reviewer for the suggestion, and we have now further investigated the relationship between isoform length and predicted protein-coding potential using multiple approaches.

First, we tested different methods to assess the relationship between isoform length and protein coding potential, finding that the positive association between the two variables is consistent even upon choosing different models to fit the data (**Reviewer Fig. 1A**, see below). Measurement of the correlation between the two variables using Spearman's index also yielded a highly significant $p\text{-value} < 2e-16$ and a $\rho = 0.54$. We additionally assessed the distribution of predicted protein-

coding probabilities upon binning isoforms by length using different methods, finding highly significant increases of coding probabilities among groups of increasing length in all cases (**Reviewer Fig 1B**, $p < 2e-16$ for each comparison).

Reviewer Figure 1. Assessing the relationship between isoform length and predicted protein-coding potential. **A.** Scatter plot displaying isoform length (x-axis) and predicted protein-coding potential (y-axis, according to CPAT) for each isoform in the novel transcriptome. The three panels display different models applied to fit the data (lm – linear model; gam – generalized additive model; loess – locally estimated scatterplot smoothing). **B.** Box plots displaying the distribution of predicted coding probabilities for isoforms grouped by length using different approaches (manually selected length bins, left; by quartile, center; by decile; right). All reported p-values are $p < 2e-16$, and were calculated using an unpaired, two-sided Wilcoxon rank sum test.

We agree with the Reviewer that a global analysis of the relationship between protein coding probability and isoform length does not offer key biological insights regarding early human development. Rather, this significance can only be meaningfully extracted on a case-by-case basis for the predictions we generated for each gene and isoform (which are accessible in our custom-built resource website). It was not our intention to make this claim, and we made sure that this is stated as such in the manuscript. However, considering the results described above, we believe this signal is statistically significant and worth reporting, as it reveals a key association between isoform length and coding probability calculated by the CPAT software (which is one of the standard and amongst the most highly cited tools for this purpose, PMID 23335781). We think this association should be made clear in order to enable users of our dataset to perform more accurate and unbiased downstream analyses from these predictions, and to better inform future experiments using the reference dataset described herein. Thus, we have added the box plot displaying the relationship between coding probability and isoform length bins to **Suppl. Figure 2A**, and have adjusted our interpretation of the data in the main text accordingly.

2. In Figure 2C, we can see the distinct patterns of isoform expression throughout all embryonic stages across all datasets, but the difference between isoform classes cannot be seen obviously from the picture.

We thank the Reviewer for the suggestion. The main purpose of **Figure 2C** is to highlight the progression of expression across developmental stages for isoforms within each class, underscoring that most novel isoform classes (most notably novel antisense and intergenic) are more highly expressed in early developmental stages, subsequently undergoing downregulation.

However, the difference between isoform classes at given stages is indeed not immediately evident. In order to make this difference clearer, we generated a new panel (now included in **Suppl. Figure 3A**) displaying the average expression of isoforms across each structural class across developmental stages for the studies spanning multiple preimplantation developmental stages (this study, Yan et al., Liu et al., Xue et al., Petropoulos et al.):

Suppl. Figure 3A. Line plots displaying average normalized gene expression levels (TPM – transcripts per million) across short-read RNA-Seq studies, grouped by isoform class and developmental stage.

This new panel clearly shows the progression of expression of isoform classes across developmental stages. These results suggest the following patterns: known isoforms are the most highly expressed out of all isoform classes, increasing in average expression in later developmental stages; NNC isoforms are the second most highly expressed category, but are slightly downregulated when progressing towards the blastocyst; NIC isoforms are approximately stably expressed throughout preimplantation development, with no evidently conserved pattern across datasets; while novel antisense and intergenic isoforms are most highly expressed prior to the 4C-8C stages, and are subsequently downregulated.

3. The authors described the TF-novel gene interaction network is “high-confidence” in line 564-565. How is it concluded?

The TF-novel gene network is generated by intersecting significant results from multiple state-of-the-art computational tools that leverage independent, orthogonal data types in order to predict putative TF-target gene interactions. To summarize:

- First, we performed a TF footprinting analysis using TOBIAS (PMID 32848148). This analysis leverages the distribution Tn5 cut sites from publicly available human preimplantation embryo ATAC-Seq data (PMID 30664750) with known TF motifs, predicting the presence of TF binding events in regions of accessible chromatin.
- To further narrow down a list of candidate TFs, we intentionally selected motifs that also display a statistically significant enrichment using HOMER (PMID 20513432).
- To additionally filter candidate TF-target pairs, we calculated predicted TF activities in each of our embryo RNA-Seq samples using VIPER. This method estimates protein activity based on the relative expression of the known targets of each TF, resulting in numeric estimates that are reported to be more robust when compared to the simple expression of the TF itself (for more details on the method, see PMID 27322546).

- The final network is generated by identifying TF-target pairs where (A) the TF is predicted to have a significant binding event in proximity of the target gene TSS, and (B) the VIPER-inferred TF activity is significantly correlated with the expression of the target gene ($p < 0.05$).

While we believe the resulting network is robust, we recognize that the term “high-confidence” is subjective and thus rephrased it to “filtered” candidates, given these tools are rigorous, but in order to avoid over interpretation.

4. In Figure. 3A, the signals of H3K4me3 and H3K27ac around novel antisense/intergenic gene TSS are weak. CAGE-seq may be helpful to validate the existence of these novel TSSs.

We thank the Reviewer for this suggestion. While the H3K4me3 and H3K27ac signals at novel antisense and intergenic TSSs are indeed not as strong as the other TSS categories, they are still significantly stronger than background, as highlighted in **Suppl. Figure 3C** ($p < 2e-16$, Wilcoxon rank-sum test). The same is true in our reanalysis of chromatin accessibility data, which displays even stronger enrichment (**Figure 3A**, top row and **Suppl. Figure 3C**, left). Nonetheless, prompted by this question alongside comments from Reviewer #3, we have now performed additional analyses of the expression patterns of novel antisense and intergenic genes to investigate possible reasons for this discrepancy. Notably, we found that many novel genes are highly expressed in oocytes (see **Figure 2C**, bottom row; **Suppl. Figures 6A-D**), and are thus likely maternally inherited. Therefore, the low levels of H3K4me3/H3K27ac/chromatin accessibility at the corresponding TSSs can likely be explained by the result that these genes are not actively transcribed in human embryos, but rather are produced in the oocyte prior to fertilization and subsequently inherited by the zygote. We have modified the text accordingly to account for this possibility.

Nonetheless, with regards to CAGE-Seq, we definitely agree that this would be a useful experiment to validate the existence of the newly identified TSSs. However, this approach could be challenging for multiple reasons. First, CAGE-Seq reads typically only span the first 20-30 nucleotides from the transcript 5' ends, which are much shorter than our Illumina RNA-Seq (125 bp paired-end) and PacBio IsoSeq reads (which often span multiple kilobases). While the limited read length is usually not an issue when investigating non-repetitive regions of the genome, it poses significant challenges when dealing with repetitive regions such as those of transposable elements (TEs), as unambiguous mapping to these regions is nearly impossible with such short reads. Since many novel genes are driven by TE promoters, CAGE would not be ideal to definitively validate the genomic location of their TSSs. Indeed, we believe that a long-read sequencing approach such as the one described herein is the ideal approach to resolve such TE-chimeric transcriptional events. Furthermore, given the limited nature of our biobank, collecting enough human embryos to generate a high-quality CAGE dataset spanning all of the profiled preimplantation stages is prohibitive from a time and resource perspective for the purpose of this project. Nonetheless, we remain interested in this approach, and we believe this would be better suited for a separate, follow-up study.

5. What is y-axis of the curves in Figure 4B, the distribution density? If the x-axis represents the percentage of short RNA-Seq reads, why the percentage may be less than 0 or more than 100%.

We thank the Reviewer for pointing this out. The y-axis in **Figure 4B** indeed represents the density of the distribution for the percentage of short RNA-Seq reads mapped to novel isoforms for each gene. The underlying data is distributed between 0 and 100%, but the smoothing function that

was applied gives the impression that the data goes beyond such boundaries. We have updated the plot by trimming the density at the true axis limits to avoid any misconceptions.

6. The cluster tree of the hierarchical clustering in some heatmaps such as in Figure 7C, D doesn't make any sense except sorting, and it is not necessary to display the meaningless cluster tree.

After consideration, we agree that the Reviewer is correct in pointing this out: we have thus removed the clustering trees from the **Figures 7C-D** as suggested.

7. The figure legend of Figure 8B, C (Line 1157-1160) need to be improved.

We have now improved the legends for the **Figures 8B** and **8C** indicated as suggested, better specifying the source of the analyzed datasets, as well as the methods used to transfer the novel isoform annotations to the respective primate genomes.

We thank the Reviewer for their valuable insight into our work and constructive feedback on how to improve the presentation of our dataset. We believe the manuscript is greatly strengthened by the novel additions.

Reviewer #3 (Remarks to the Author):

Torre et al present a valuable resource of isoform-resolved transcriptome of the human preimplantation embryo. The data clearly highlights the underestimated complexity of the human transcriptome at the earliest stages of development especially during the period from zygote to zygotic genome activation. I therefore recommend publication following minor revision.

We thank the Reviewer for appreciating the importance and novelty of our work, and are excited to get this data in the hands of the community as an updated transcriptome for use in human developmental studies and beyond.

The novel isoforms detected during early human embryo may be maternally loaded transcripts. It would be interesting to specifically examine this in previously published datasets of oocytes. doi: 10.1038/ncomms9207 contain a large number of oocytes and may be a useful dataset to include for such analysis.

We do agree that analyzing also (metaphase II) mature oocytes is valuable, especially when focusing on maternally expressed genes. Unfortunately, we did not have access to oocyte

samples for IsoSeq data generation in our study, since freezing human oocytes for fertility preservation was available many years after embryo freezing. Therefore, female patients who froze oocytes in our unit still store them for future fertility use, and they are not donated for research. However, since we do agree that integration of published oocyte data is a valuable addition, we have included the appropriate additional analyses below.

Specifically, we downloaded, processed, and quantified known and novel isoform expression using short-read bulk RNA-Seq datasets from two published studies that have sequenced a large number of human oocytes and zygotes: Töhönen et al. (PMID 26360614, as recommended above), and Asami et al. (PMID 34936886). Due to the large number of replicates and presence of both oocyte and zygote samples, these datasets are well suited to investigate whether the newly identified isoforms are maternally loaded, and present in embryos following fertilization.

The analysis revealed that a large fraction of novel isoforms are already detected in human oocytes across integrated datasets from Liu et al. and Asami et al. Thus, this suggests that many of the newly identified isoforms and genes are already expressed in the oocyte, and likely maternally loaded, as correctly speculated by the Reviewer, and we clarified this as a valuable discussion point. The results of this analysis have been included in the revised **Figure 2C** shown here:

Figure 2C. Percentage of isoforms in each developmental stage, grouped by isoform class and average expression level across short-read RNA-Seq datasets (TPM – Transcript Per Million).

We have additionally expanded the analysis of oocyte expression in the context of **Figure 7**, where we had identified five clusters of novel genes with different expression patterns across the 1C-blastocyst stages in our samples. Many genes annotated in the “Early cluster” and “2C cluster” appear to be already detected in oocyte samples in the previously integrated short-read RNA-Seq datasets across human embryos (**Suppl. Figure 6A**). Investigating expression patterns at corresponding syntenic genomic locations in published primate embryo RNA-Seq datasets also

reveals a remarkable degree of concordance, especially for “Early cluster” and “2C cluster” gene expression in macaque oocytes and early embryos (**Suppl. Figure 6B**):

Suppl. Figure 6A. Box plots displaying normalized gene expression values (VST) of novel gene clusters in short-read RNA-Seq data from three integrated publicly available embryo short-read RNA-Seq studies. **6B.** Percentage of genes in each cluster that map to the macaque and marmoset genomes, and are expressed across developmental stages in corresponding preimplantation short-read RNA-Seq datasets.

We have now further investigated the expression of these novel gene clusters in the context of the extensive oocyte RNA-Seq data from Tohonen et al. and Asami et al. mentioned above. As we now show in the new **Suppl. Figure 6C**, genes from the “Early cluster” and “2C cluster” are broadly expressed in oocytes from both datasets, suggesting these are largely maternally inherited, while other gene clusters are predominantly unexpressed, suggesting instead that they are activated during or after EGA, in concordance with patterns observed in our and other RNA-Seq studies spanning multiple preimplantation stages:

Suppl. Figure 6C. Box plots displaying normalized counts (DESeq2 size factors) of novel gene clusters in oocyte short-read RNA-Seq data from Asami et al and Töhönen et al.

In Figure 2 D I'm not sure the Mazid et al. integration is performed in the correct manner as the primed cells looks like they are mixing with the TE cells of Petropoulos et al. Also, the 8CLC cluster with E4 cells of Petropoulos et al and not with the E3.

We thank the Reviewer for the important comment regarding the scRNAseq data integration.

Indeed, the Reviewer is correct in pointing out that the 8CLC cells from Mazid et al. cluster primarily with E4 cells from the Petropoulos et al. dataset. However, close inspection of the equivalent analysis performed by Mazid et al. in their article initially reporting 8CLCs (PMID 35314832, **Figure 1D**, see below) also reveals that, while some sorted 8CLCs (light green triangles, see below) indeed cluster with E3 cells from Petropoulos et al. (red squares), the majority of 8CLCs cluster more closely to E4 cells (dark orange squares), as well as morula-stage cells from Yan et al. (see just below the region highlighted in the box).

Mazid et al. (PMID 35314832), **Figure 1D**. UMAP comparing the human E7 to E3 stage, human ES cell passage 10 (hESC P10) to 8C embryo with primed ES cells, 4CL-D12 naive ES cells, stepwise e4CL-D5 cells and sorted 8CLCs from stepwise e4CL-D5 cells profiled using SMART-seq2.

Furthermore, the primed ESCs also display some degree of mixing among E7-stage Petropoulos cells. Overall, because the appearance of UMAP plots is highly sensitive to the choice of parameters and data integration method, and given that the primary purpose of our **Figure 2** is to highlight the detection of novel isoforms in published RNA-Seq studies, rather than comparing existing studies, we decided to replace this manuscript panel. We have now included other panels that we believe are better suited to highlight the expression of novel isoforms across existing datasets and human embryonic stages (see below – this also addresses a comment regarding this panel by Reviewer #2).

How significant is the increased gene set score for the Mazid et al, considering it is -0.1 in the 8CLC which is lower than in the E3 in Petropoulos et al? Could the low score be explained by the possibility that many of these transcripts are maternally loaded?

The Reviewer brings up a very interesting question with regards to novel gene expression in 8CLCs. The scores displayed in the box plots were calculated independently for each scRNA-seq dataset, comparing the normalized expression of gene sets of interest (i.e. novel antisense and intergenic) to the aggregated expression of control feature sets (using the AddModuleScore function from the Seurat package, PMID 34062119). While the low gene set scores may indeed be consistent with the maternal loading of these transcripts, interpretation of the specific values is likely not the ideal approach to address this important question, especially because these scores are calculated separately for each dataset prior to data integration, making direct comparison between these distributions challenging.

Following this feedback, and to further address a comment by Reviewer #2, we have now restructured the analysis in the following ways: 1) we replaced the panel with ones that more quantitatively display the detection of novel genes across datasets, and 2) added new analyses (in the context of **Figure 5**) to further investigate whether the novel genes are maternally inherited, and whether there are discrepancies between the sets of novel genes expressed in early stage embryos (from Petropoulos et al.) and 8CLCs (from Mazid et al.).

First, we have updated the bar plots in **Figures 2C** to display the percentage of genes detected across stages from the Petropoulos et al. RNA-Seq data (see below). Furthermore, we now display similar results for each cluster in the Mazid et al. (8CLC) and Kagawa et al. (human blastoid) RNA-Seq datasets in **Figure 2D**:

Figure 2C. Percentage of isoforms in each developmental stage, grouped by isoform class and average expression level across short-read RNA-Seq datasets profiling human preimplantation embryos and oocytes (TPM – Transcript Per Million). **2D.** As above, but displaying data from single-cell short-read RNA-Seq datasets (SmartSeq2) profiling in-vitro models of human preimplantation development (8CLCs and blastoids).

The results indicate that, while novel antisense and intergenic genes are widely expressed in E3-stage embryos (Petropoulos et al.), detection of these genes is lower in human 8CLCs (Mazid et al.), and even lower in primed PSCs and in cells from in-vitro derived human blastoids (Kagawa et al.). These results are suggestive of the fact that many such genes are maternally inherited, and not reactivated in in-vitro models for EGA such as 8CLCs.

To further investigate this possibility, we analyzed the expression of the five novel gene clusters which we had previously identified in **Figure 7A** across these integrated single-cell RNA-Seq datasets. Notably, we found that E3-stage embryos from Petropoulos et al. display broad expression of early- and 2C-cluster novel genes (new **Suppl. Figure 6D**, see below), which we showed to be highly expressed in human oocytes and thus maternally inherited (**Suppl. Figure 6C**, shown above). By contrast, 8CLCs do not broadly express genes in these clusters, primarily reactivating genes in the late- and 8C- clusters instead:

Suppl. Figure 6D. Box plots displaying normalized expression (Transcripts per Million - TPM) of novel gene clusters in short-read RNA-Seq data from Mazid et al. and Petropoulos et al.

Together, these results show that a large fraction of novel genes are maternally inherited and not reactivated in 8CLCs. This is consistent with what the Reviewer initially suggested, and we believe this is ultimately a better approach to highlight this difference.

Could also the relatively “silent” epigenetic signatures for “Novel antisense gene TSS” and “Novel intergenic gene TSS” in Figure 3A, B also be explained by maternal loading without active transcription at later stages?

We thank the Reviewer for bringing this up. This is certainly possible, and indeed in light of the results discussed above, we believe this may be the case. We have updated the main manuscript text to account for this possibility.

We thank the Reviewer for the thoughtful feedback on the manuscript and for the helpful insights on how to ameliorate it. We believe the integration of oocyte RNA-Seq data and additional suggestions have added significant value and discussion context to our manuscript.

REVIEWERS' COMMENTS

Reviewer #2 (Remarks to the Author):

The authors have mostly answered the issues I raised in the earlier review. I think this revised version has been improved largely. I recommend the publication of this revised version.

Reviewer #3 (Remarks to the Author):

The authors have satisfyingly addressed all my original comments and I believe the manuscript is acceptable for publication.